# Incentivized Truthful Communication for Federated Bandits

**Zhepei Wei**[†*]  **Chuanhao Li**[†*]  **Tianze Ren**[†]  **Haifeng Xu**[‡]  **Hongning Wang**[†]
[†]University of Virginia      [‡]University of Chicago
{tqf5qb, cl5ev, tr2bx, hw5x}@virginia.edu  haifengxu@chicago.edu

## Abstract

To enhance the efficiency and practicality of federated bandit learning, recent advances have introduced incentives to motivate communication among clients, where a client participates only when the incentive offered by the server outweighs its participation cost. However, existing incentive mechanisms naively assume the clients are truthful: they all report their true cost and thus the higher cost one participating client claims, the more the server has to pay. Therefore, such mechanisms are vulnerable to strategic clients aiming to optimize their own utility by misreporting. To address this issue, we propose an incentive compatible (i.e., truthful) communication protocol, named TRUTH-FEDBAN, where the incentive for each participant is independent of its self-reported cost, and reporting the true cost is the only way to achieve the best utility. More importantly, TRUTH-FEDBAN still guarantees the sub-linear regret and communication cost without any overhead. In other words, the core conceptual contribution of this paper is, for the first time, demonstrating the possibility of simultaneously achieving incentive compatibility and nearly optimal regret in federated bandit learning. Extensive numerical studies further validate the effectiveness of our proposed solution.

## 1 Introduction

Bandit learning (Lattimore & Szepesvári, 2020) addresses the exploration-exploitation dilemma in interactive environments, where the learner repeatedly chooses actions and observes the corresponding rewards from the environment. Subject to different goals of the learner, e.g., maximizing cumulative rewards (Abbasi-Yadkori et al., 2011; Auer et al., 2002) vs., identifying the best arm (Audibert et al., 2010; Garivier & Kaufmann, 2016), bandit algorithms have been widely applied in various real-world applications, such as model selection (Maron & Moore, 1993), recommender systems (Li et al., 2010a;b), and clinical trials (Durand et al., 2018). Most recently, propelled by the increasing scales of data across various sources and public concerns about data privacy, there has been growing research effort devoted to federated bandit learning, which enables collective bandit learning among distributed learners while preserving data privacy of each learner. Recent advances in this line of research mainly focus on addressing the communication bottleneck in the federated network, which leads to communication-efficient protocols for both non-contextual (Landgren et al., 2016; Martínez-Rubio et al., 2019; Shi et al., 2020; Zhu et al., 2021) and contextual bandits (Wang et al., 2020; Huang et al., 2021; Li et al., 2022; 2023) under various environment settings.

However, almost all previous works assume clients are altruistic in sharing their local data with the server whenever communication is triggered (Wang et al., 2020; Li & Wang, 2022a; He et al., 2022). This limits their practical deployment in real-world scenarios involving *individual rational* clients who share data only if provided with clear benefits. The only notable exception is Wei et al. (2023), where incentive is provided to motivate client's participation in federated learning. Nevertheless, their protocol naively assumes the clients are truthful in reporting their participation cost; and thus, they simply calculate incentives by each client's claimed cost, leaving it as a design flaw for strategic clients to exploit. Therefore, how to design an *incentive compatible* mechanism for federated bandits that ensures truthful reporting while still preserving the near-optimal regret and communication cost still remains an open research problem.

---

[*]Equal Contribution

| Method | Regret | Communication Cost | *IR* | *IC* | *SC* |
|---|---|---|:---:|:---:|:---:|
| `DisLinUCB` (Wang et al., 2020) | $O(d\sqrt{T}\log T)$ | $O(N^2 d^3 \log T)$ | ✗ | ✗ | ✗ |
| `Inc-FedUCB` (Wei et al., 2023) | $O(d\sqrt{T}\log T)$ | $O(N^2 d^3 \log T)$ | ✓ | ✗ | ✗ |
| `Truth-FedBan` (Our Algorithm 1) | $O(d\sqrt{T}\log T)$ | $O(N^2 d^3 \log T)$ | ✓ | ✓ | ✓ |

Table 1: Comparison with related works, where *IR*, *IC* and *SC* represent the guarantee of *individual rationality*, *incentive compatibility*, and *social cost near-optimality*, respectively.

Following Wei et al. (2023)'s setting for learning contextual linear bandits in a federated environment, we develop the first incentive compatible communication protocol TRUTH-FEDBAN, which ensures the clients can only achieve their best utility by reporting the true participation costs. Specifically, instead of simply paying a client by its claimed cost, we decouple the calculation of incentive from the target client's reported cost, while preserving individual rationality through a *critical-value* based payment design that depends on all other clients' report cost. Besides the theoretical guarantee on truthfulness, we also empirically demonstrate that misreporting cost brings no benefit to the client's utility. More encouragingly, we prove that this can be achieved without any compromise in maintaining the near-optimal performance in regret and communication cost.

On the other hand, in addition to the above desiderata, maintaining a minimal *social cost* is also an important objective in the incentivized communication problem, especially in practical applications. Following classical economic literature (Procaccia & Tennenholtz, 2013), social cost is defined as the sum of true participation costs among all participating clients. While incentivizing all clients' participation ensures nearly optimal performance (Wang et al., 2020), it can be scientifically trivial (e.g., paying everyone to have all of them participate) and practically undesirable — it not only brings unnecessary burden for the server, but can also expose unnecessary clients to potential downsides of participation (e.g., privacy breaches, added resource consumption, etc.), resulting in worse social cost. Minimizing social cost while ensuring sufficient client participation is non-trivial, as it in nature is NP-hard (see Eq. (1)). Though the method proposed by Wei et al. (2023) achieves sub-linear regret and communication cost (albeit assuming truthfulness), it provides no guarantee on the social cost. In contrast, our proposed TRUTH-FEDBAN guarantees both sub-linear regret and near-optimal social cost, with only a constant-factor approximation ratio. To better illustrate our contribution, we compare the proposed TRUTH-FEDBAN with the most related works in Table 1.

## 2 RELATED WORK

### 2.1 FEDERATED BANDIT LEARNING

Federated bandit learning has been well investigated for sequential decision making in distributed environments. These studies mainly differ in how they model the clients' and environment characteristics, which can be categorized into 1) *bandit-wise*: problem profile (e.g., context-free (Martínez-Rubio et al., 2019; Shi & Shen, 2021; Shi et al., 2020) vs. contextual (Wang et al., 2020)) and decision set (e.g., fixed (Huang et al., 2021) vs. time-varying (Li & Wang, 2022b)), and 2) *system-wise*: client type (e.g., homogeneous (He et al., 2022) vs. heterogeneous (Li & Wang, 2022a)), network type (e.g., peer-to-peer (P2P) (Dubey & Pentland, 2020) vs. star-shaped (Wang et al., 2020)), and communication type (e.g., synchronous (Li et al., 2022) vs. asynchronous (Li et al., 2023)).

Most recently, Wei et al. (2023) expand this spectrum by introducing the notion of incentivized communication, where the server has to pay the clients for their participation. Despite being free from the long-standing assumption about the client's willingness of participation in literature, they still assume truthfulness of clients in cost reporting. Specifically, their incentive calculation is based on the client's self-reported cost, which leads to serious vulnerability in adversarial scenarios as clients can exploit this flaw, ultimately paralyzing the federated learning system. This is particularly concerning in real-world applications where self-interested clients are motivated to strategically game the system for increased utilities, i.e., increase the difference between incentives offered by the server and actual participation costs. Our work aims to address this issue by introducing a truthful incentive mechanism under which clients reporting true costs is in their best interest, while ensuring near-optimal learning performance.

## 2.2 MECHANISM DESIGN

Mechanism design (Nisan & Ronen, 1999) has been playing a crucial role in the fields of economics, computer science and operation research, with fruitful auction-like real-world applications such as matching markets (Roth, 1986), resource allocation (Procaccia, 2013), online advertisement pricing (Aggarwal et al., 2006). Typically, the auctioneer (server) aims to sell/purchase one or more entries of a collection to/from multiple bidders (clients), with the objective of maximizing social welfare or minimizing social cost. The goal of mechanism design is to incentivize clients to truthfully report the values of the entries (i.e., *truthfulness*), while ensuring non-negative utilities if they participate in the mechanism (i.e., *individual rationality*).

The Vickrey-Clarke-Groves (VCG) mechanism (Vickrey, 1961; Clarke, 1971; Groves, 1973) is probably the most well-known truthful mechanism. Despite having been well explored in many theoretical studies, VCG is rarely applied in practical applications due to its computational inefficiency. This is because VCG requires finding an optimal solution to the concerned problem, which is often NP-hard (Archer & Tardos, 2001). Otherwise, truthfulness cannot be guaranteed when VCG mechanisms are applied to sub-optimal solutions (Lehmann et al., 2002). To facilitate study on this issue, Mu'Alem & Nisan (2008) identified the key character of a truthful mechanism and reduced the problem to designing a monotone algorithm (see Section 4.1). One notable recent related work is (Kandasamy et al., 2023), where the authors model repeated auctions as a bandit learning problem for the server, with clients being unaware of their values but able to provide bandit feedback based on the server's allocation. The server's goal is to find allocations that maximize social welfare, while ensuring the clients' truthfulness in their feedback. In contrast, in our work, clients know their participation costs and are concerned to solve the bandit problem collectively. The server's goal is to incentivize clients' participation for regret minimization, while ensuring the clients' truthfulness in cost reporting and minimizing social cost.

In terms of problem formulation, our work is closest to the *hiring-a-team* task in procurement auctions (Talwar, 2003; Archer & Tardos, 2007), where the server aims to incentivize a set of self-interested clients to jointly perform a task. One standard assumption in this task is that the environment is monopoly-free, i.e., no single client exists in all feasible sets (Iwasaki et al., 2007). The reason is that if a client is essential, it has the bargaining power to ask for infinite incentive. In this paper, we do not assume a monopoly-free environment, otherwise additional environment assumptions will be needed (e.g., how the context or arms should distribute across clients). Instead, we are intrigued in studying the origin and impact of the monopoly issue from both theoretical and empirical perspectives. And we also rigorously prove that we can eliminate the issue via hyper-parameter control in our mechanism (see Lemma 7).

### 2.3 MECHANISM DESIGN IN FEDERATED LEARNING

On the other hand, there have been growing efforts in investigating mechanism design in the context of *federated learning* (Pei, 2020; Tu et al., 2022). For example, Karimireddy et al. (2022) introduced a contract-theory based incentive mechanism to maximize data sharing while avoiding free-riding clients. In their design, every client gets different snapshots of the global model with different levels of accuracy as incentive, and truthfully reporting their data sharing costs is the best response under the proposed incentive mechanism. Therefore, there is no overall performance guarantee and their focus is on investigating the level of accuracy the system can achieve under this truthful incentive mechanism. Le et al. (2021) also investigated truthful mechanism design in the application scenario of wireless communication, where server's goal is to maximize the system's social welfare, with respect to a knapsack upper bound constraint. In contrast, in our problem the server is obligated to improve the overall performance of the learning system, i.e., obtaining near-optimal regret among all clients. Furthermore, our optimization problem (defined in Eq. (3)) aims at minimizing the social cost, with respect to a submodular lower bound constraint. Therefore, despite we share a similar idea of using the monotone participant selection rule and critical-value based payment design to guarantee truthfulness, the underlying fundamental optimization problems are completely different, and consequently their solution cannot be used to solve our problem. Besides pursuing the truthfulness guarantee in mechanism design under the collaborative/federated setting, the other related line of research focuses on designing incentive mechanisms that ensures fairness among distributed clients (Blum et al., 2021; Xu et al., 2021; Sim et al., 2020; Donahue & Kleinberg, 2023), which is also an important direction, despite being beyond the scope of our work. To our best knowledge, our work is the first attempt that studies truthful mechanism design for federated bandit learning.

## 3 PRELIMINARY: INCENTIVIZED FEDERATED BANDITS

In this section, we present the incentivized communication problem for federated bandits in general and the existing solution framework under the linear reward assumption (Wang et al., 2020). More precisely, we focus our discussions on the learning objectives, including minimizing regret, communication cost, social cost, and ensuring truthfulness.

Consider a learning system with 1) $N$ distributed *strategic* and *individual rational* clients that repeatedly interact with the environment by taking actions to receive rewards, and 2) a central server responsible for motivating the clients to participate in federated learning via incentives. As in line with Wei et al. (2023), we assume the clients can only communicate with the server, forming a star-shaped communication network. Specifically, at each time step $t \in [T]$, an arbitrary client $i_t \in [N]$ chooses an arm $\mathbf{x}_t \in \mathcal{A}_t$ from its given arm set $\mathcal{A}_t \subseteq \mathbb{R}^d$. Then, client $i_t$ receives a reward $y_t = \mathbf{x}_t^\top \theta_\star + \eta_t \in \mathbb{R}$, where $\theta_\star$ is the unknown parameter shared by all clients and $\eta_t$ denotes zero-mean sub-Gaussian noise. Typically, in the centralized setting of bandit learning, a ridge regression estimator $\hat{\theta}_t = V_{g,t}^{-1} b_{g,t}$ is constructed for arm selection based on the sufficient statistics from all $N$ clients at time step $t$, where $V_{g,t} = \sum_{s=1}^t \mathbf{x}_s \mathbf{x}_s^\top$ and $b_{g,t} = \sum_{s=1}^t \mathbf{x}_s y_s$. In contrast, since communication does not occur at every time step $t$ in the federated setting, each client $i$ only has a delayed copy of $V_{g,t}$ and $b_{g,t}$, denoted as $V_{i,t} = V_{g,t_{\text{last}}} + \Delta V_{i,t}, b_{i,t} = b_{g,t_{\text{last}}} + \Delta b_{i,t}$, where $V_{g,t_{\text{last}}}, b_{g,t_{\text{last}}}$ are the aggregated statistics shared by the server in the last communication, and $\Delta V_{i,t}, \Delta b_{i,t}$ are the accumulated local updates that client $i$ has collected from the environment since $t_{\text{last}}$.

**Regret and Communication Cost**  One key objective of the learning system is to minimize the (pseudo) regret for all $N$ clients across the entire time horizon $T$, i.e., $R_T = \sum_{t=1}^T r_t$, where $r_t = \max_{\mathbf{x} \in \mathcal{A}_t} \mathbf{E}[y|\mathbf{x}] - \mathbf{E}[y_t|\mathbf{x}_t]$ is the instantaneous regret of client $i_t$ at time step $t$. Meanwhile, a low communication cost is also desired to keep the efficiency of federated learning, which is measured by the total number of scalars transferred throughout the system up to time $T$. Intuitively, more frequent communication leads to lower regret. For example, communicating at every time step recovers the centralized setting, leading to the lowest regret, but with an undesirably high communication cost. Efficient communication protocol design becomes the key to balance regret and communication cost. And using determinant ratio to measure the outdatedness of the sufficient statistics stored on the server side against those on the client side has become the reference solution to control communication in federated linear bandits (Wang et al., 2020; Li & Wang, 2022a).

**Incentivized Communication**  When dealing with individual rational clients, additional treatment is needed to facilitate communication, as it becomes possible that no client participates unless properly incentivized thus leading to terrible regret. In other words, client $i$ only participates if its utility $u_{i,t} = \mathcal{I}_{i,t} - D_{i,t}$ is non-negative, where $\mathcal{I}_{i,t}$ is the server-provided incentive, and $D_{i,t}$ is the client's participation cost. To address this challenge and maintain near-optimal learning outcome, Wei et al. (2023) pinpointed the core optimization problem in incentivized communication as follows:

$$\min_{S_t \in 2^{\widetilde{S}}} \sum_{i \in S_t} \widehat{D}_{i,t} \quad s.t. \quad \frac{\det(V_{g,t}(S_t))}{\det(V_{g,t}(\widetilde{S}))} \geq \beta \tag{1}$$

where $\widehat{D}_{i,t}$ is client $i$'s reported participation cost, $S_t$ is the set of clients selected to participate at time step $t$, $\widetilde{S} = \{1, 2, \cdots, N\}$ is the set of all clients, $\beta$ is specified as an input to the algorithm, and $V_{g,t}(S) = V_{g,t_{\text{last}}} + \sum_{j \in S} \Delta V_{j,t}$. In particular, they assume the clients' reported cost is simply the true cost, i.e., $\widehat{D}_{i,t} = D_{i,t}$. A heuristic search algorithm is executed to solve the optimization problem whenever the standard communication event (Wang et al., 2020) is triggered. A detailed description of this communication protocol is provided in Appendix F.

Note that Wei et al. (2023)'s work is limited to a constant cost setting of $D_{i,t} = C_i \cdot \mathbb{I}(\Delta V_{i,t} \neq \mathbf{0})$, which restricts the actual cost $D_{i,t}$ of client $i$ to be independent of time and its local updates $\Delta V_{i,t}$. In our work, we relax it to $D_{i,t} = f(\Delta V_{i,t})$, where $f$ can be any reasonable data valuation function, and even time-varying[1]. Moreover, their proposed solution for Eq. (1) fails to provide any approximation guarantee on the objective, thus having no guarantee on the social cost. Below, we provide a formal definition of *truthfulness* and *social cost* employed in this paper.

---

[1]In fact, our proposed TRUTH-FEDBAN works with any realization of the valuation function, as all that matters is that client $i$ has a value $D_{i,t}$ for its data at time step $t$.

**Definition 1 (Truthfulness)** *An incentive mechanism is truthful (i.e., incentive compatible) if at any time $t$ the utility $u_{i,t}$ of any client $i$ is maximized when it reports its true participation cost, i.e., $\widehat{D}_{i,t} = D_{i,t}$, regardless of the reported costs of the other clients' $\widehat{D}_{-i,t}$.*

**Definition 2 (Social Cost)** *The social cost of the learning system is defined as the total actual costs incurred by all participating clients in the incentivized client set $S_t$, i.e., $\sum_{i \in S_t} D_{i,t}$.*

Note that the social cost defined above is different from the incentive cost studied in Wei et al. (2023), which is the total payment the server made to all clients. As truthfulness is assumed in their setting, the payment that the server needs to make to incentivize a client is trivially upper bounded by the client's true cost. However, in order to ensure truthfulness in our setting, the server needs to overpay the selected clients (compared with their true cost). In the case where there exists monopoly client as introduced in Section 2.2, an infinite incentive cost is required.

## 4 METHODOLOGY

### 4.1 CHARACTERIZATION OF TRUTHFUL INCENTIVE MECHANISMS

Our idea stems from the seminal result of Mu'Alem & Nisan (2008), who provided a characterization of a truthful incentive mechanism as a combination of a monotone selection rule and a critical value payment scheme, which reduces the problem of designing a truthful mechanism to that of designing a monotone selection rule. Though it is originally intended for combinatorial auctions in economics, we are the first to extend it to the incentivized communication problem in federated bandit learning, laying the foundations for future work.

**Definition 3 (Monotonicity)** *The selection rule for the set $S_t$ is monotone if for any client $i$ and any reported costs of the other clients $\widehat{D}_{-i,t}$, client $i$ will remain selected whenever it reports $\widehat{D}'_{i,t} \leq \widehat{D}_{i,t}$, provided it is incentivized when reporting $\widehat{D}_{i,t}$.*

Furthermore, according to Mu'Alem & Nisan (2008), any monotone selection rule of the incentive mechanism has an associated critical payment scheme, with its definition given below.

**Definition 4 (Critical Payment)** *Let $\mathcal{M}$ be a monotone selection rule of the incentive mechanism and $S_t$ be the set of selected clients, then for any client $i$ and any reported costs of the other clients $\widehat{D}_{-i,t}$, there exists a critical value $c_{i,t}(\mathcal{M}, \widehat{D}_{-i,t}) \in (\mathbb{R}_+ \cup \infty)$ such that $i \in S_t$, $\forall \widehat{D}_{i,t} < c_{i,t}(\mathcal{M}, \widehat{D}_{-i,t})$, and $i \notin S_t$, $\forall \widehat{D}_{i,t} > c_{i,t}(\mathcal{M}, \widehat{D}_{-i,t})$.*

In this way, we can decouple the incentive $\mathcal{I}_{i,t}$ for client $i$ from its reported participation cost $\widehat{D}_{i,t}$, and calculate the critical value based only on the other clients' reported costs $\widehat{D}_{-i,t}$. Formally,

$$\mathcal{I}_{i,t} = c_{i,t}(\mathcal{M}, \widehat{D}_{-i,t}) \cdot \mathbb{I}(i \in S_t) \tag{2}$$

which is fundamentally different from the incentive design in (Wei et al., 2023) where $\mathcal{I}_{i,t} = \widehat{D}_{i,t} \cdot \mathbb{I}(i \in S_t)$, as our payment method leaves no room for strategic clients to manipulate the incentive and benefit from misreporting.

### 4.2 TRUTH-FEDBAN: A TRUTHFUL MECHANISM FOR INCENTIVIZED COMMUNICATION

To balance regret and communication cost, while ensuring truthfulness and minimizing social cost, our proposed incentive mechanism TRUTH-FEDBAN inherits the incentivized communication protocol by Wei et al. (2023), with the distinction in implementing a truthful incentive search. As stated above, the truthfulness of clients is ensured once we devise a monotone algorithm for client selection, combined with a critical payment scheme. But straightforward monotone algorithms (e.g., a greedy algorithm ranking clients by their claimed costs) offer no guarantee on social cost. To address this challenge, we rewrite the original optimization problem in Eq. (1) into the following equivalent submodular set cover (SSC) problem, where $g(S)$ is a submodular set function (see Definition 11).

$$\min_{S_t \in 2^{\widetilde{S}}} \sum_{i \in S_t} \widehat{D}_{i,t} \ \ s.t. \ \ g_t(S_t) \geq \log \beta, g_t(S_t) = \log \frac{\det(V_{g,t}(S_t))}{\det(V_{g,t}(\widetilde{S}))} \tag{3}$$

---

**Algorithm 1** Truthful Incentive Search

**Require:** $\beta, \epsilon > 0$
1: $S_t \leftarrow \emptyset, \widetilde{S} = \{1, 2, \cdots, N\}$
2: $b \leftarrow \min_{i \in \widetilde{S}} \widehat{D}_{i,t}$
3: **while** $g_t(S_t) < (1 - e^{-1}) \log \beta$ **do**
4: $\quad b \leftarrow (1 + \epsilon)b$
5: $\quad S_t \leftarrow \text{GREEDY}(\widetilde{S}, b)$
6: **return** $S_t$

---

**Algorithm 2** GREEDY

**Require:** $\widetilde{S}, b$
1: $S_t \leftarrow \emptyset$
2: **while** $\sum_{i \in S_t} \widehat{D}_{i,t} < b$ **do**
3: $\quad u \leftarrow \underset{j \in \widetilde{S} \setminus S_t : \widehat{D}_{j,t} + \sum_{i \in S_t} \widehat{D}_{i,t} < b}{\arg\max} \frac{g_t(S_t \cup \{j\}) - g_t(S_t)}{\widehat{D}_{j,t}}$
4: $\quad S_t \leftarrow S_t \cup \{u\}$
5: **return** $S_t$

---

Inspired by Iyer & Bilmes (2013), we propose Algorithm 1 that achieves a constant-factor bi-criteria approximation for both the objective and constraint in the problem defined by Eq. (3). As outlined above, we first initialize a minimal budget (Line 2) for social cost and repeatedly increase the budget (Line 4) until the resulting client set found by Algorithm 2 satisfies the specified condition (Line 3).

In Algorithm 2, for a given budget $b$, we iteratively find the best set of clients from the complete client set until the budget cannot afford more clients. At each iteration, all the remaining non-selected clients are ranked based on their contribution-to-cost ratio (and hence being greedy). The algorithm then chooses the client with the highest ratio while ensuring the total cost of all selected clients is within the budget (Line 3 of Algorithm 2). The correctness of our method hinges on the following crucial monotonicity property that we prove. Interestingly, despite the wide use of greedy algorithms in submodular maximization, this monotonicity result is unknown in previous literature to the best of our knowledge. We thus present it as a proposition in case it is of independent interest.

**Proposition 5 (Monotonicity)** *Algorithm 1 is monotone.*

It is not difficult to show that Algorithm 2 is monotone for a fixed input budget $b$ — that is, if client $i$ is selected under $\widehat{D}_{i,t}$ by Algorithm 2, it remains selected when it reports any $\widehat{D}'_{i,t} \leq \widehat{D}_{i,t}$. But it is highly non-trivial to prove monotonicity for Algorithm 1. This is because decreasing a client's reported cost can cause a different output by Algorithm 2 and, consequently, terminate the search process in Algorithm 1 at a different budget $b$ with a potentially different selection of participant set $S_t$. We prove Proposition 5 by showing the resulting objective value $g_t(S_t)$ from Algorithm 2's selection of clients is non-decreasing with respect to its input budget $b$. The proof is a bit involving since Algorithm 2 is an approximate algorithm and generally outputs sub-optimal solutions. We will have to show that the quality of these sub-optimal solutions — which can be close to or far away from the exact optimality — will not degenerate as the budget $b$ increases. The proof of the above property, together with the formal proof of Proposition 5, can be found in Appendix A.

**Lemma 6** *If the selection rule of a truthful mechanism is computable in polynomial time, so is the critical payment scheme (Mu'Alem & Nisan, 2008).*

Note that in the star-shaped communication network, only the server has the necessary information to calculate the critical value of each client, and we assume the server is committed not to tricking the clients. Due to space limit, we leave the detailed critical payment calculation to Appendix G. In particular, as we do not assume a monopoly-free environment, a client's critical value could be infinite at a certain point, as introduced in Section 2.2. Nonetheless, Lemma 7 shows that this infinite payment issue can be essentially eliminated by hyper-parameter control. The time complexity analysis of Algorithm 1 can be found in Appendix H.

**Lemma 7 (Elimination of Infinite Critical Value)** *With parameter $\beta \leq (1 + tL^2/\lambda d)^{-d}$ in Algorithm 1, no client will be essential in any communication round at time step $t$.*

A detailed proof is provided in Appendix D. Building upon the properties above, we are now ready to state the main incentive guarantee of our TRUTH-FEDBAN protocol.

**Theorem 8** *The incentive mechanism induced by Algorithm 1 is (a) **truthful** in the sense that every client achieves the highest utility by reporting its true participation cost; and (b) **individually rational** in the sense that every client's utility of participating in the mechanism is non-negative.*

**Proof** The truthfulness guarantee directly follows Lemma 5. Below, we further elaborate on the impact of misreporting. Denote $S_t$ and $S'_t$ as the participant sets when client $i$ truthfully reports and misreports its data sharing cost as $\widehat{D}_{i,t} = D_{i,t}$ and $\widehat{D}'_{i,t} \neq D_{i,t}$, respectively. Let $c_{i,t}$ be the critical value of client $i$, and $u_{i,t}$ and $u'_{i,t}$ be its utilities in the above two conditions respectively. According to Definition 4, we have $i \in S_t$ whenever $\widehat{D}_{i,t} < c_{i,t}$, and $i \notin S'_t$ whenever $\widehat{D}'_{i,t} > c_{i,t}$. Moreover, if $i \notin S_t$, then $u_{i,t} = 0$. For simplicity, the subscript $t$ is omitted in the following discussion. Specifically, there are four possible cases: 1) $i \in S$ and $i \in S'$, as critical payment is independent from the client's reported cost $\widehat{D}_i$ and $\widehat{D}'_i$, therefore $u'_i - u_i = (c_i - D_i) - (c_i - D_i) = 0$; 2) $i \in S$ and $i \notin S'$, in this case, $\widehat{D}_i = D_i < c_i < \widehat{D}'_i$. Therefore, $u'_i - u_i = 0 - (c_i - D_i) = D_i - c_i < 0$; 3) $i \notin S$ and $i \in S'$, in this case, $\widehat{D}'_i < c_i < \widehat{D}_i = D_i$. Therefore, $u'_i - u_i = (c_i - D_i) - 0 < 0$; 4) $i \notin S$ and $i \notin S'$, in this case, both utilizes are zero, therefore $u'_i - u_i = 0 - 0 = 0$. To conclude, there is no benefit to misreport under our truthful mechanism design in all cases, and only reporting the true data sharing cost can lead to the client's best utility.

We now prove the individual rationality. Given the truthfulness guarantee, each client $i$ reports its true cost $\widehat{D}_{i,t} = D_{i,t}$, and only gets incentivized if $\widehat{D}_{i,t} < c_{i,t}$. Therefore, the utility of client $i$ is $u_{i,t} = c_{i,t} - D_{i,t} > 0$ if client $i$ gets incentivized; otherwise, $u_{i,t} = 0$. In either case, client $i$ is ensured to have a non-negative utility, which completes the proof. ∎

## 4.3 Learning Performance of Truth-FedBan Protocol

The truthfulness property above helps the system induce desirable clients participation behaviors. In this subsection, we demonstrate the learning performance of TRUTH-FEDBAN under these client behaviors. Our main results are the following guarantees regarding total social cost that the TRUTH-FEDBAN protocol has to suffer and the resultant regret guarantee it induces.

**Theorem 9 (Social Cost)** *For any $\epsilon > 0$, using Algorithm 2 to search for participants in Algorithm 1 provides a $[1 + \epsilon, 1 - e^{-1}]$ bi-criteria approximation solution for the problem defined in Eq. (3). In other words, to maintain a social cost that is within a $(1 + \epsilon)$ factor of the optimal value, it necessitates a relaxation of the constraint by a factor of $(1 - e^{-1})$. Formally,*

$$\sum_{i \in S_t} \widehat{D}_{i,t} \leq (1 + \epsilon) \sum_{i \in S_t^\star} \widehat{D}_{i,t} \ \text{ and } \ g_t(S_t) \geq (1 - e^{-1}) \log \beta$$

*where $S_t$ is the output of Algorithm 1, and $S_t^\star$ is the ground-truth optimizer of Eq. (3).*

**Proof** Denote the optimal objective value of Eq. (3) as OPT. For the solution $S_t^\star$, we have OPT $= \sum_{i \in S_t^\star} \widehat{D}_{i,t}$ and $g_t(S_t^\star) \geq \log \beta$. To simplify out discussions, we omit the subscript $t$ and let $S_b$ and $b$ be the output set and terminating budget of Algorithm 1 for solving the problem in Eq. (3), and $S_{b'}$ and $b' = b/(1 + \epsilon)$ be the set and budget at the previous iteration before termination, then we have

$$\begin{cases} g(S_{b'}) < (1 - e^{-1}) \log \beta & (4) \\ g(S_b) \geq (1 - e^{-1}) \log \beta & (5) \end{cases}$$

Denote $S_{b'}^\star$ as the optimal solution for the subroutine search problem with budget $b'$ (denote the problem solved by Algorithm 2 in Line 5 of Algorithm 1 as SUBPROBLEM). According to Sviridenko (2004), the approximation ratio of Algorithm 2 for this SUBPROBLEM is $(1 - e^{-1})$, i.e.,

$$g(S_{b'}) \geq (1 - e^{-1}) g(S_{b'}^\star) \tag{6}$$

Combining Eq. (4) and Eq. (6), we have $g(S_{b'}^\star) < \log \beta$. Furthermore, we can show that OPT $> b'$ by contradiction. Assuming OPT $\leq b'$, then $S^\star$ is a feasible solution for the SUBPROBLEM, and thus $g(S^\star) \leq g(S_{b'}^\star) < \log \beta$. However, this contradicts the fact that $g(S^\star) \geq \log \beta$, so OPT $> b'$. Hence, we can show that the objective value of solution $S_b$ satisfies the following inequality:

$$\sum_{i \in S_b} \widehat{D}_i \leq b = (1 + \epsilon) b' < (1 + \epsilon) \text{OPT} \tag{7}$$

This, combined with Eq. (5), concludes the proof. ∎

Since $\widehat{D}_{i,t} = D_{i,t}$ is guaranteed (see Theorem 8), Theorem 9 directly bounds the social cost as defined in Definition 2. Note that as indicated by Theorem 9, we can flexibly choose any desired level of social cost by adjusting the parameter $\epsilon$, which allows us to accommodate various computation resources in practical scenarios. For example, in the case where computation is not a limiting factor and the core objective is to minimize the social cost, we can set the factor $(1 + \epsilon)$ to be almost 1, approaching the optimal social cost. Moreover, though this bi-criteria approximation slightly deviates from the constraint of the original problem in Eq. (3), it only incurs a constant-factor gap of $(1 - e^{-1})$, and Theorem 10 shows that we still attain near-optimal regret and communication cost, despite this deviation (see proof in Appendix E).

**Theorem 10 (Regret and Communication Cost)** *Under threshold $\beta$, with high probability the communication cost of* TRUTH-FEDBAN *satisfies* $C_T = O(Nd^2) \cdot P = O(N^2d^3 \log T)$, *where* $P = O(Nd \log T)$ *is the total number of communication rounds, under the communication threshold* $D_c = \frac{T}{N^2 d \log T} - \sqrt{\frac{T^2}{N^2 dR \log T}} \log \beta^{(1-e^{-1})}$ *in Algorithm 6, where* $R = \lceil d \log(1 + \frac{T}{\lambda d}) \rceil$. *Furthermore, by setting* $\beta^{(1-e^{-1})} \geq e^{-\frac{1}{N}}$, *the cumulative regret is* $R_T = O\left(d\sqrt{T} \log T\right)$.

## 5 EXPERIMENTS

To validate our solution, we create a simulated federated bandit learning environment with context feature dimension $d = 5$ and $N = 25$ clients sequentially interacting with the environment for a fixed time horizon $T$. Due to space limit, more implementation details can be found in Appendix G. The results, averaged over 5 runs, are presented alongside the standard deviation.

### 5.1 COMPARISON BETWEEN DIFFERENT TRUTHFUL INCENTIVE MECHANISMS

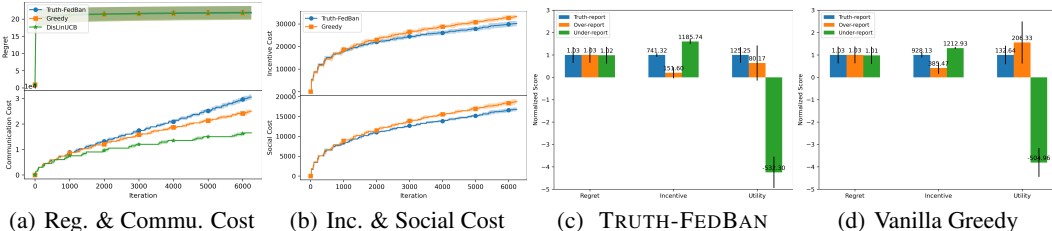

| (a) Reg. & Commu. Cost | (b) Inc. & Social Cost | (c) TRUTH-FEDBAN | (d) Vanilla Greedy |

Figure 1: Comparison between TRUTH-FEDBAN and vanilla greedy incentive mechanism.

We compare TRUTH-FEDBAN with a vanilla greedy algorithm (Algorithm 3). Despite Algorithm 3 also induces a monotone mechanism (and thus truthful), it does not admit any constant-factor approximation guarantee, hence is less theoretically exciting compared to TRUTH-FEDBAN. A comprehensive analysis regrading this baseline method can be found in Appendix B. As reported in Figure 1(a) and Figure 1(b), TRUTH-FEDBAN achieved competitive sub-linear regret and communication cost compared to DisLinUCB (Wang et al., 2020), with lower incentive and social costs compared to the baseline greedy method, validating our theoretical analysis.

### 5.2 IMPACT ON MISREPORTING

**Micro-level Study.** In this experiment, we study how misreporting affects an individual, in terms of the client's regret, incentive and utility. To do so, we randomly designate a client to keep misreporting throughout the entire time horizon while keeping the others being truthful, and compare the corresponding outcome for this client. We take *truth-report* as the benchmark and plot the individual's total regret, incentive, and utility on the same chart using the normalized score (the respective value divided by that under truth-report), along with the actual value on top of each bar. As presented in Figure 1(c) and Figure 1(d), both TRUTH-FEDBAN and the greedy method demonstrate the ability to prevent client from benefiting via misreporting. It is important that the incentive payment (i.e., critical value) for the client is subject to the incentive mechanism and independent of its claimed cost. Therefore, though under-reporting may encourage the client to be selected by the incentive mechanism, its net utility essentially becomes negative. Meanwhile, despite over-reporting cost only undertakes the risk of being ruled out and losing incentives, it is surprising that this behav-

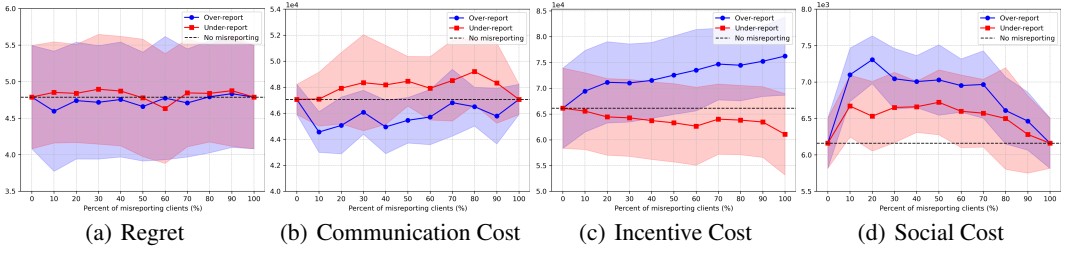

Figure 2: Overall impact of misreporting.

ior leads to a slightly higher utility under the vanilla greedy incentive mechanism. We attribute this to the reduced participation cost incurred by the client — the less it participates, the less it suffers.

**Macro-level Study.** As presented in Figure 2, we also empirically investigate how different levels of misreporting across the set of clients affect the entire federated learning system. Specifically, we vary the number of misreporting clients from 0% to 100% to investigate the impact on overall system performance, including regret, communication cost, incentive cost, and social cost. Generally, as guaranteed by our communication protocol, the overall regret under different degrees of misreporting remains virtually identical to the situation when no client misreports. Meanwhile, the communication cost tends to increase when clients under report and decrease when they over report. This aligns with our algorithm's design, which selects clients based on their value-to-cost ratio (Line 3 in Algorithm 2). For example, when a client under reports, its ratio increases, which increases its chance to be selected in communication, hence leading to an increased communication cost.

An interesting finding that might seem contradictory to our discovery in the previous micro-level study is the overall impact of over reporting on incentive costs, which implies that the more clients over report, the higher the incentives they will receive. But our finding in the micro-level study suggests that over reporting brings no benefit to the client's individual utility under TRUTH-FEDBAN. We note that the observation in our macro-level study is due to *collusion* among clients — once a sufficient group of clients colludes, the server has to increase the critical value or even pay infinity. This actually rationalizes individual client's commitment to be truthful, as they are unaware of others clients' decision on truthfulness. Meanwhile, this finding reveals the vulnerability of the incentivized truthful communication to collusion, leaving an interesting avenue for future work to explore. On the other hand, it can be observed that both overreporting and underreporting hurt the social cost until the misreporting ratio reaches approximately 50%. This is interesting from the perspective of societal divisions — when the society is equally divided into two parts, the social cost is at its largest. And as division decreases, the cost becomes lower. For example, when the misreporting ratio reaches 100%, meaning that everyone in the system is misreporting, the social cost resets the scenario where no one misreports, marking the establishment of a new stability in the system.

## 6 CONCLUSION

In this work, we introduce the first truthful incentivized communication protocol TRUTH-FEDBAN for federated bandit learning, where a set of strategic and individual rational clients are incentivized to truthfully report their cost to participate distributed learning. Our key contribution is to design a monotone client selection rule and its corresponding critical value based payment scheme. We establish the theoretical foundations for incentivized truthful communication, under which not only the social cost but also the regret and communication cost obtain their near-optimal performance. Numerical simulations verify our theoretical results, especially the truthfulness guarantee, i.e., individual clients' utility can only be maximized when reporting their true cost.

Our work opens a broad new direction for future exploration. First of all, our truthful incentivized communication protocol is not only limited to federated bandit learning, but can be applied to general distributed learning environments where self-interested clients need to be incentivized for collaborative learning. Second, our truthful guarantee is proved for every round of communication, but it is unclear whether a client can do long-term planning to game the system. For example, keep over reporting until it becomes monopoly, ultimately leading to an infinite incentive for its participation. Last but not least, although we no longer assume clients are truthful, we still assume they are not malicious, i.e., they only want to maximize their own utility. In practice, it is necessary to investigate the problem under an adversarial context, e.g., malicious clients intentionally misreport their costs to hurt other clients' utilities or system's learning outcome.

ACKNOWLEDGMENTS

We thank the anonymous reviewers for their insightful and constructive comments. This project is partially supported by NSF Award IIS-2213700 and IIS-2128019. Haifeng Xu is supported by an NSF Award CCF-2303372, an Army Research Office Award W911NF-23-1-0030, an Office of Naval Research Award N00014-23-1-2802, and AI2050 program at Schmidt Sciences (Grant G-24-66104).

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

## A  PROOF OF MONOTONICITY (PROPOSITION 5)

Our proof of monotonicity relies on the submodularity property and the following lemma. Note that our proof holds true for any time step $t$, and thus the subscript $t$ is omitted below to keep our notations simple.

**Definition 11 (Submodularity)** *A set function* $g : 2^S \to \mathbb{R}$ *is submodular, if for every* $A \subseteq B \subseteq S$ *and* $i \in S \setminus B$ *it holds that*

$$g(A \cup \{i\}) - g(A) \geq g(B \cup \{i\}) - g(B)$$

**Lemma 12** *Increasing the input budget of Algorithm 2 always leads to a no worse output. Formally, denote the output of Algorithm 2 as* $S_b$ *and* $S_{b'}$ *under different input budgets* $b$ *and* $b'$. *For any budget pair* $b' > b$, *we must have either* $S_{b'} = S_b$ *or* $g(S_{b'}) > g(S_b)$.

*Proof of Lemma 12.* Considering two input budget $b$ and $b'$ and the corresponding outputs $S_b$ and $S_{b'}$ of Algorithm 2. In the following, we show that $g(S_{b'}) \geq g(S_b)$ if $b' > b$. Without loss of generality, we denote $S_b = \{j_1, j_2, \cdots, j_n\} \in 2^{\widetilde{S}}$ and $S_{b'} = \{k_1, k_2, \cdots, k_m\} \in 2^{\widetilde{S}}$ as the corresponding output, where $\widetilde{S} = \{1, 2, \cdots, N\}, 1 \leq n \leq N, 1 \leq m \leq N$.

Under different budget, the selected client in each round can vary due to the changed constraint in Line 3 of Algorithm 2. For example, under budget $b$, a client with the largest ratio may not be selected because including it would cause the total cost to exceed $b$. In contrast, under budget $b'$, it can be selected due to the increased budget. Consequently, this can create different output sequences $S_b$ and $S_{b'}$.

Let $\tau$ be the first time when the two sequences diverge, i.e, $j_i = k_i, \forall 1 \leq i < \tau$, and $j_\tau \neq k_\tau$. If such a $\tau$ does not exist, the two sequences are precisely the same and we will have $S_{b'} = S_b$. In the remainder of this proof, we assume $\tau$ exists and show that $g(S_{b'}) > g(S_b)$ consequently. Let $S_b^\tau$ and $S_{b'}^\tau$ be the set that contains the first $\tau$ elements in $S_b$ and $S_{b'}$, so we have $S_b^{\tau-1} = S_{b'}^{\tau-1}$. According to the greedy strategy (Line 3 of Algorithm 2), it is clear that $\widehat{D}_{k_\tau} > \sum_{i=\tau}^n \widehat{D}_{j_i}$, meaning that the cost of client $k_\tau$ is even higher than the total costs of clients in $S_b \setminus S_b^{\tau-1}$, which is the reason that $k_\tau$ appears in the output $S_{b'}$ under a larger budget $b'$ but is not (thus is skipped) in the output $S_b$ under $b$. Moreover, this also implies that client $k_\tau$ has a larger value-to-cost ratio than that of any client in $S_b \setminus S_b^{\tau-1}$ at the $\tau$-th round, formally

$$\frac{g(S_{b'}^{\tau-1} \cup \{k_\tau\}) - g(S_{b'}^{\tau-1})}{\widehat{D}_{k_\tau}} \geq \frac{g(S_b^{\tau-1} \cup \{j_i\}) - g(S_b^{\tau-1})}{\widehat{D}_{j_i}}, \forall i \in [\tau, n] \tag{8}$$

For clarity, we denote the value of client $k_\tau$ as $v(k_\tau | S_{b'}^{\tau-1}) = g(S_{b'}^{\tau-1} \cup \{k_\tau\}) - g(S_{b'}^{\tau-1})$, quantifying how much client $k_\tau$ can improve the objective function $g$ with respect to the set $S_{b'}^{\tau-1}$. Then we have

$$\sum_{i=\tau}^n \frac{v(j_i | S_b^{\tau-1})}{\widehat{D}_{j_i}} \cdot \widehat{D}_{j_i} \leq \frac{v(k_\tau | S_{b'}^{\tau-1})}{\widehat{D}_{k_\tau}} \sum_{i=\tau}^n \cdot \widehat{D}_{j_i} < \frac{v(k_\tau | S_{b'}^{\tau-1})}{\widehat{D}_{k_\tau}} \cdot \widehat{D}_{k_\tau} = v(k_\tau | S_{b'}^{\tau-1})$$

where the first inequality follows from Eq. (8), and the second one holds true because $\widehat{D}_{k_\tau} > \sum_{i=\tau}^n \widehat{D}_{j_i}$. Therefore, we can derive that

$$v(k_\tau | S_{b'}^{\tau-1}) > \sum_{i=\tau}^n v(j_i | S_b^{\tau-1}) \tag{9}$$

Now we are ready to compare the objective value of $S_b$ and $S_{b'}$, and show that $g(S_{b'}) > g(S_b)$. By simple decomposition, we can rewrite $g(S_b)$ as follows

$$\begin{aligned}
g(S_b) &= g(\{j_1, j_2, \cdots, j_n\}) \\
&= g(\emptyset) + [g(\{j_1\}) - g(\emptyset)] + [g(\{j_1, j_2\}) - g(j_1)] + \cdots + [g(S_b) - g(S_b \setminus \{j_n\})] \\
&= g(\emptyset) + v(j_1 | S_b^0) + v(j_2 | S_b^1) + \cdots + v(j_n | S_b^{n-1}) \\
&= g(\emptyset) + \sum_{i=1}^\tau v(j_i | S_b^{i-1}) + \sum_{p=\tau}^n v(j_p | S_b^{p-1})
\end{aligned}$$

Likewise, we have

$$g(S_{b'}) = g(\emptyset) + \sum_{i=1}^{\tau} v(k_i|S_{b'}^{i-1}) + \sum_{p=\tau}^{n} v(k_p|S_{b'}^{p-1})$$

Recall that $j_i = k_i, \forall i < \tau$, and thus we have $v(j_i|S_b^{i-1}) = v(k_i|S_{b'}^{i-1}), \forall i < \tau$. Therefore,

$$g(S_{b'}) - g(S_b) = \sum_{i=\tau}^{n} v(k_i|S_{b'}^{i-1}) - \sum_{i=\tau}^{n} v(j_i|S_b^{i-1})$$

$$= v(k_\tau|S_{b'}^{\tau-1}) + \sum_{i=\tau+1}^{n} V(k_i|S_{b'}^{i-1}) - \sum_{i=\tau}^{n} v(j_i|S_b^{i-1})$$

$$> \sum_{i=\tau}^{n} v(j_i|S_b^{\tau-1}) - \sum_{i=\tau}^{n} v(j_i|S_b^{i-1}) + \sum_{i=\tau+1}^{n} v(k_i|S_{b'}^{i-1})$$

$$> \sum_{i=\tau}^{n} v(j_i|S_b^{\tau-1}) - v(j_i|S_b^{i-1})$$

$$> 0$$

where the first inequality directly follows Eq. (9), and the last step utilizes the submodularity property (see Definition 11) of the submodular function $g$, i.e., $v(j_i|S_b^{\tau-1}) > v(j_i|S_b^{i-1}), \forall i > \tau$. This concludes the proof. ∎

Now we are ready to prove the monotonicity of Algorithm 1 by contradiction.

*Proof of Proposition 5.* An algorithm is monotone if a client $\alpha$ remains selected by the algorithm whenever its reported cost satisfies $\widehat{D}'_\alpha < \widehat{D}_\alpha$, provided it gets selected when reporting $\widehat{D}_\alpha$. Let $S = \{i_1, i_2, \cdots, i_n\}$ and $b$ be the resulting participant set and budget determined by Algorithm 1 when client $\alpha$ reports $\widehat{D}_\alpha$. Without loss of generality, we set $\alpha = i_k$, where $1 \leq k \leq n$, and denote $S^k = \{i_1, i_2, \cdots, i_k\}$ as the set of clients selected before $\alpha$. According to the greedy selection strategy in Algorithm 2, we have

$$\frac{g(S^{k-1} \cup \{\alpha\}) - g(S^{k-1})}{\widehat{D}_\alpha} > \frac{g(S^{k-1} \cup \{i\}) - g(S^{k-1})}{\widehat{D}_i}, \forall i \in \widetilde{S} \setminus S^k : \widehat{D}_i + \sum_{j \in S^{k-1}} \widehat{D}_j \leq b \quad (10)$$

Denote $S'$ and $b'$ as the resulting participant set and budget determined by Algorithm 1 when client $\alpha$ reports $\widehat{D}'_\alpha < \widehat{D}_\alpha$. Since decreasing client $\alpha$'s claimed cost will increase the ratio in the left-hand side of Eq. (10), it will remain selected (no later than the $k$-th round) when $b' \leq b$, otherwise the terminating participant set $S_{b'}$ is not sufficient. The algorithm only deviates from this when the following condition is true:

$$\frac{g(S^{k-1} \cup \{\alpha\}) - g(S^{k-1})}{\widehat{D}'_\alpha} < \frac{g(S^{k-1} \cup \{i\}) - g(S^{k-1})}{\widehat{D}_i}, \exists i \in \widetilde{S} \setminus S^k : \widehat{D}_i + \sum_{j \in S^{k-1}} \widehat{D}_j \leq b'$$

According to Eq. (10), this is only possible when $b' > b$ because the increased budget allows additional candidate clients with both larger value and cost, potentially surpassing the largest affordable ratio under $b$. However, it contradicts the fact that any feasible terminating budget must be at most $b$ — as Lemma 12 guarantees that a larger budget input to Algorithm 2 must always result in either exactly the same set or a different set with strictly higher objective value. Meanwhile, the terminating condition (Line 3 of Algorithm 1) ensures that the entire search process will promptly terminate once it finds the minimum budget that satisfies the constraint. Therefore, given budget $b$ already satisfies the constraint, it is impossible for the algorithm to terminate with a solution that has a higher budget than $b$, which finishes the proof. ∎

## B    GREEDY INCENTIVE SEARCH

In contrast to Algorithm 1, one straightforward alternative is to adopt the vanilla greedy method to solve the problem in Eq. (3), as presented in Algorithm 3. The idea is to iteratively rank all non-selected clients according to their individual value-to-cost ratio and choose the one with the largest ratio (Line 3-4), until the resulting participant set satisfies the constraint (Line 2).

---

**Algorithm 3** Vanilla Greedy Incentive Search

---

1: $S_t \leftarrow \emptyset, \widetilde{S} = \{1, 2, \cdots, N\}$
2: **while** $g_t(S_t) < \log \beta$ **do**
3: $\quad i \leftarrow \arg\max_{j \in \widetilde{S} \setminus S_t} \frac{g_t(S_t \cup \{j\}) - g(S_t)}{\widehat{D}_{j,t}}$
4: $\quad S_t \leftarrow S_t \cup \{i\}$
5: **return** $S$.

---

It is not difficult to verify this straightforward greedy algorithm is also monotonic, as decreasing a client's claimed cost essentially encourages its selection, thus making it a truthful mechanism. One notable difference between this greedy incentive search algorithm and our truthful incentive search (Algorithm 1) is that it does not compromise for the constraint. As a result, as pointed out by previous studies (Wolsey, 1982), this greedy algorithm does not admit any constant-factor approximation guarantee (i.e., it becomes problem instance specific), as shown in Lemma 13.

**Lemma 13 (Theorem 2 of Wolsey (1982))** *Under parameter $\beta$ and clients' reported participation cost $\widehat{D}_t = \{\widehat{D}_{1,t}, \cdots, \widehat{D}_{N,t}\}$, Algorithm 3 is guaranteed to obtain a participant set $S_t$ such that*

$$\sum_{i \in S} \widehat{D}_{i,t} \leq (1 + \ln \min\{\lambda_1, \lambda_2, \lambda_3\}) \sum_{i \in S_t^\star} \widehat{D}_{i,t} \ \text{ and } \ g_t(S_t) \geq \log \beta$$

*in which $\lambda_1 = \max_{i,k}\{\frac{g_t(\{i\}) - g_t(\emptyset)}{g_t(S_t^k \cup \{i\}) - g_t(S_t^k)} \mid g_t(S_t^k \cup \{i\}) - g_t(S_t^k) > 0\}$ where the denominator is the smallest non-zero marginal gain from adding any element $i \in \widetilde{S}$ to the intermediate set $S_t^k$, i.e., the set contains the first $k$ elements of the output set $S_t$, and the numerator is the largest singleton value of $g$; $\lambda_2 = \frac{\sigma_1}{\sigma_K}$ where $K$ is the total number of iterations in the greedy search and $\sigma_k = \max_i \frac{g_t(S_t^k \cup \{i\}) - g_t(S_t^k)}{\widehat{D}_{i,t}}$; $\lambda_3 = \frac{g(\widetilde{S}) - g(\emptyset)}{g(\widetilde{S}) - g(S_t^{K-1})}$.*

Alternatively, we can reformulate Algorithm 3 into an equivalent counterpart (Algorithm 4) that provides a bi-criteria approximation guarantee similar to Algorithm 1. Note that these two variants essentially lead to the same outcome when parameterized with $\beta_1 = \beta^{(1-e^{-1})}$ and $\beta_2 = \beta$, where $\beta_1$ and $\beta_2$ are the specified hyper-parameters in Algorithm 3 and Algorithm 4, respectively.

---

**Algorithm 4** Greedy Incentive Search (V2)

---

**Require:** $\beta, \widetilde{S} = \{1, 2, \ldots, N\}$
1: $B \leftarrow \text{ORDEREDBUDGET}(\widetilde{S})$
2: $S_t \leftarrow \emptyset, b \leftarrow 0, k \leftarrow 0$
3: **while** $g_t(S_t) < (1 - e^{-1}) \log \beta$ **do**
4: $\quad b \leftarrow b + B[k]$
5: $\quad S \leftarrow \text{GREEDY}(\widetilde{S}, b)$
6: $\quad k \leftarrow k + 1$
7: **return** $S_t$

---

**Algorithm 5** ORDEREDBUDGET

---

**Require:** $\widetilde{S} = \{1, 2, \cdots, N\}$
1: $S_t \leftarrow \emptyset, B \leftarrow \emptyset$
2: **while** $\widetilde{S} \setminus S_t \neq \emptyset$ **do**
3: $\quad u \leftarrow \arg\max_{j \in \widetilde{S} \setminus S_t} \frac{g_t(S_t \cup \{j\}) - g_t(S_t)}{\widehat{D}_{j,t}}$
4: $\quad S_t \leftarrow S_t \cup \{u\}$
5: $\quad B \leftarrow B \cup \{\widehat{D}_{u,t}\}$
6: **return** $B$

---

**Lemma 14** *Under parameter $\beta$ and clients' reported participation cost $\widehat{D}_t = \{\widehat{D}_{1,t}, \cdots, \widehat{D}_{N,t}\}$, Algorithm 4 provides a bi-criteria approximation such that*

$$\sum_{i \in S_t} \widehat{D}_{i,t} \leq \max \widehat{D}_t + \sum_{i \in S_t^\star} \widehat{D}_{i,t} \ \text{ and } \ g_t(S_t) \geq (1 - e^{-1}) \log \beta$$

*where $S_t$ is the output of Algorithm 1, and $S_t^\star$ is the ground-truth optimizer of problem defined in Eq. (1).*

*Proof of Lemma 14.* The proof of this lemma largely repeats that of Lemma 9, with a minor difference in Eq. (7) (i.e., $b = (1 + \epsilon)b'$ vs., $b = b' + B[k]$). Unlike Algorithm 1 slightly increasing the

budget by a constant factor $(1 + \epsilon)$, Algorithm 4 increases the budget in a pre-ordered way based on the result of Algorithm 5. Similar to Eq. (7), the subscript $t$ is omitted, and we have

$$\sum_{i \in S_b} \widehat{D}_i \leq b = b' + B[k] \leq \max \widehat{D} + \sum_{i \in S^\star} \widehat{D}_i \tag{11}$$

Additionally, it is not difficult to see $g_t(S_t) \geq (1 - e^{-1}) \log \beta$ since this is the terminating condition of Algorithm 4. Combining both completes the proof. ∎

## C   Technical Lemmas

**Lemma 15 (Lemma 10 of Abbasi-Yadkori et al. (2011))** *Suppose* $\mathbf{x}_1, \mathbf{x}_2, \cdots, \mathbf{x}_t \in \mathbb{R}^d$ *and for any* $1 \leq s \leq t$, $\|\mathbf{x}_s\|_2 \leq L$. *Let* $\overline{V}_t = \lambda I + \sum_{s=1}^t \mathbf{x}_s \mathbf{x}_s^\top$ *for some* $\lambda > 0$. *Then,*

$$\det(\overline{V}_t) \leq (\lambda + tL^2/d)^d.$$

**Lemma 16 (Lemma 11 of Abbasi-Yadkori et al. (2011))** *Let* $\{X_t\}_{t=1}^\infty$ *be a sequence in* $\mathbb{R}^d$, $V$ *is a* $d \times d$ *positive definite matrix and define* $V_t = V + \sum_{s=1}^t X_s X_s^\top$. *Then we have that*

$$\log \left( \frac{\det(V_n)}{\det(V)} \right) \leq \sum_{t=1}^n \|X_t\|_{V_{t-1}^{-1}}^2.$$

*Further, if* $\|X_t\|_2 \leq L$ *for all t, then*

$$\sum_{t=1}^n \min \left\{ 1, \|X_t\|_{V_{t-1}^{-1}}^2 \right\} \leq 2 \left( \log \det(V_n) - \log \det V \right) \leq 2 \left( d \log \left( \left( \mathrm{trace}(V) + nL^2 \right) /d \right) - \log \det V \right).$$

## D   Proof of Lemma 7

Our proof utilizes the following matrix determinant lemma (Harville, 2008).

**Lemma 17 (Matrix Determinant Lemma)** *Let* $A \in \mathbb{R}^{n \times n}$ *be an invertible n-by-n matrix, and* $B, C \in \mathbb{R}^{n \times m}$ *are n-by-m matrices, we have that*

$$\det(A + BC^\top) = \det(A) \det(I_m + C^\top A^{-1} B)$$

*Proof of Lemma 7.* It is known that the infinite critical value is unavoidable for a monopoly client under the truthful mechanism design. To eliminate this issue, we first analyze the root cause of the existence of a monopoly client. Denote $\widetilde{V}_t$ as the covariance matrix constructed by all sufficient statistics available in the system at time step $t$, and $\Delta V_{i,t} = X_n^\top X_n$, $X_n \in \mathbb{R}^{\Delta t \times d}$. Specifically, client $i$ is a monopoly, i.e., being essential to satisfy the constraint in Eq. (3) at time step $t$, such that having all the other $N - 1$ clients' data still cannot satisfy the constraint. According to Lemma 17, plugging in $A = \widetilde{V}_t - \Delta V_{i,t}$ and $B = C = X_n^\top$, we have

$$\frac{\det(\widetilde{V}_t - \Delta V_{i,t})}{\det(\widetilde{V}_t)} = \frac{1}{\det(I_{\Delta t} + X_n(\widetilde{V}_t - \Delta V_{i,t})^{-1} X_n^\top)}$$

where $\Delta V_{i,t} = X_n^\top X_n$, $X_n \in \mathbb{R}^{\Delta t \times d}$, $\Delta t$ represents the number of new data points in $\Delta V_{i,t}$. Next, we show that there exists a lower bound of the ratio above, such that as long as we set the hyper-parameter $\beta$ less than the lower bound, it is guaranteed that no client can be essential. Moreover, for a positive definite matrix $A \in \mathbb{R}^{d \times d}$, we have $A^{-1} \preccurlyeq \frac{I}{\lambda_{min}(A)}$ where $\lambda_{min}(A)$ denotes the minimum eigenvalue of $A$. Plugging in $A = \widetilde{V}_t - \Delta V_{i,t}$, we have $(\widetilde{V}_t - \Delta V_{i,t})^{-1} \preccurlyeq \frac{I}{\lambda_{min}(\widetilde{V}_t - \Delta V_{i,t})} \preccurlyeq \frac{I}{\lambda}$,

where $\lambda > 0$ is the regularization parameter defined in Eq. (12). It follows that

$$
\frac{1}{\det(I_{\Delta t} + X_n(\widetilde{V}_t - \Delta V_{i,t})^{-1}X_n^\top)} \geq \frac{1}{\det(I_{\Delta t} + \frac{1}{\lambda}X_n X_n^\top)}
$$

$$
= \frac{1}{\det(I_d + \frac{1}{\lambda}X_n^\top X_n)} = \frac{\lambda^d}{\det(\lambda I_d + \Delta V_{i,t})}
$$

$$
\geq \frac{\lambda^d}{\det(V_{i,t} + \lambda I_d)}
$$

$$
\geq \frac{\lambda^d}{(\lambda + tL^2/d)^d} = (1 + tL^2/\lambda d)^{-d}
$$

where the second step holds by elementary algebra, the third step utilizes the fact that $V_{i,t} \succcurlyeq \Delta V_{i,t}$, and the last step follows from Lemma 15. Therefore, as long as we set $\beta \leq (1 + tL^2/\lambda d)^{-d}$, it is guaranteed that no client will be essential at time step $t$. This finishes the proof. ∎

## E  COMMUNICATION COST AND REGRET ANALYSIS

As TRUTH-FEDBAN directly inherits from the basic protocol proposed in (Wei et al., 2023) with a truthful incentive mechanism, most part of the proof for communication cost and regret analysis (Theorem 4) in their paper extends to our problem setting. Therefore, with slight modifications, we can achieve the same sub-linear guarantee.

In essence, the only difference in terms of establishing the theoretical bounds for regret and communication cost between our method and (Wei et al., 2023) lies in the relaxation of the constraint in Eq. (3), which deviated from the original constraint in Eq. (1) by a constant-factor gap of $(1-e^{-1})$. Moreover, as we reformulate the determinant ratio constraint (i.e., $\frac{\det(V_{g,t}(S_t))}{\det(V_{g,t}(\widetilde{S}))} \geq \beta$) into a log determinant ratio constraint (i.e., $\log\frac{\det(V_{g,t}(S_t))}{\det(V_{g,t}(\widetilde{S}))} \geq (1 - e^{-1})\log\beta$), the notion of $\beta$ in our work is slightly different from that in their work. Specifically, denote the hyper-parameter in their method as $\overline{\beta}$, then any $\overline{\beta}$ used in their theoretical results can be replaced by our notation of $\beta$ via the transformation $\overline{\beta} = \beta^{1-e^{-1}}$.

In the following, we present the corresponding theoretical results of our proposed TRUTH-FEDBAN and refer the readers to the proof details in Theorem 4 of (Wei et al., 2023).

**Lemma 18 (Communication Frequency Bound)** *By setting the communication threshold* $D_c = \frac{T}{N^2 d \log T} - (1-e^{-1})\sqrt{\frac{T^2}{N^2 dR \log T}}\log\beta$, *the total number of communication rounds is upper bounded by*

$$
P = O(Nd\log T)
$$

*where* $R = \lceil d\log(1 + \frac{T}{\lambda d})\rceil = O(d\log T)$.

**Communication Cost:** In each communication round, all clients first upload $O(d^2)$ scalars to the server and then download $O(d^2)$ scalars. According to Lemma 18, the total communication cost is $C_T = P \cdot O(Nd^2) = O(N^2 d^3 \log T)$.

**Lemma 19 (Instantaneous Regret Bound)** *Given parameter* $\beta$, *with probability* $1 - \delta$, *the instantaneous pseudo-regret* $r_t = \langle\theta^\star, \mathbf{x}^\star - \mathbf{x}_t\rangle$ *in $j$-th communication round is bounded by*

$$
r_t = O\left(\sqrt{d\log\frac{T}{\delta}}\right) \cdot \|\mathbf{x}_t\|_{\widetilde{V}_{t-1}^{-1}} \cdot \sqrt{\frac{1}{\beta^{(1-e^{-1})}} \cdot \frac{\det(V_{g,t_j})}{\det(V_{g,t_{j-1}})}}
$$

*Proof of Theorem 10.* We followed the notion of *good epoch* and *bad epoch* defined in (Wang et al., 2020). Combining with Lemma 16, we can bound the accumulative regret in the good epochs as,

$$
REG_{good} = O\left(\frac{d}{\sqrt{\beta^{1-e^{-1}}}} \cdot \sqrt{T} \cdot \sqrt{\log\frac{T}{\delta} \cdot logT}\right).
$$

Furthermore, we can show that the regret across all bad epochs satisfies,

$$REG_{bad} = O\left(Nd^{1.5}\sqrt{D_c \cdot \log \frac{T}{\delta}} \log T\right).$$

Using the communication threshold $D_c = \frac{T}{N^2 d \log T} - (1 - e^{-1})\sqrt{\frac{T^2}{N^2 dR \log T}} \log \beta$ specified in Lemma 18, we have

$$R_T = REG_{good} + REG_{bad}$$

$$= O\left(\frac{d}{\sqrt{\beta^{1-e^{-1}}}}\sqrt{T}\log T\right) + O\left(Nd^{1.5}\log^{1.5} T \cdot \sqrt{\frac{T}{N^2 d \log T} + \frac{T}{Nd \log T} \log \frac{1}{\beta^{1-e^{-1}}}}\right)$$

Henceforth, by setting $\beta^{1-e^{-1}} > e^{-\frac{1}{N}}$, we can show that $\frac{T}{N^2 d \log T} > \frac{T}{Nd \log T} \log \frac{1}{\beta^{1-e^{-1}}}$, and therefore

$$R_T = O\left(\frac{d}{\sqrt{\beta^{1-e^{-1}}}}\sqrt{T}\log T\right) + O\left(d\sqrt{T}\log T\right) = O\left(d\sqrt{T}\log T\right)$$

This concludes the proof. ∎

## F GENERAL FRAMEWORK FOR INCENTIVIZED FEDERATED BANDITS

---

**Algorithm 6** Incentivized Communication for Federated Linear Bandits

---

**Require:** $D_c \geq 0$, $\widehat{D}_t = \{\widehat{D}_{1,t}, \cdots, \widehat{D}_{N,t}\}$, $\sigma$, $\lambda > 0$, $\delta \in (0, 1)$
1: Initialize: **[Server]** $V_{g,0} = \mathbf{0}_{d \times d} \in \mathbb{R}^{d \times d}$, $b_{g,0} = \mathbf{0}_d \in \mathbb{R}^d$
2:             $\Delta V_{-j,0} = \mathbf{0}_{d \times d}, \Delta b_{-j,0} = \mathbf{0}_d, \forall j \in [N]$
3:      **[All clients]** $V_{i,0} = \mathbf{0}_{d \times d}, b_{i,0} = \mathbf{0}_d, \Delta V_{i,0} = \mathbf{0}_{d \times d}, \Delta b_{i,0} = \mathbf{0}_d, \Delta t_{i,0} = 0, \forall i \in [N]$
4: **for** $t = 1, 2, \ldots, T$ **do**
5:      **[Client $i_t$]** Observe arm set $\mathcal{A}_t$
6:      **[Client $i_t$]** Select arm $\mathbf{x}_t \in \mathcal{A}_t$ by Eq. (12) and observe reward $y_t$
7:      **[Client $i_t$]** Update: $V_{i_t,t} \mathrel{+}= \mathbf{x}_t \mathbf{x}_t^\top$, $b_{i_t,t} \mathrel{+}= \mathbf{x}_t y_t$
8:             $\Delta V_{i_t,t} \mathrel{+}= \mathbf{x}_t \mathbf{x}_t^\top$, $\Delta b_{i_t,t} \mathrel{+}= \mathbf{x}_t y_t$, $\Delta t_{i_t,t} \mathrel{+}= 1$
9:      **if** $\Delta t_{i_t,t} \log \frac{\det(V_{i_t,t} + \lambda I)}{\det(V_{i_t,t} - \Delta V_{i_t,t} + \lambda I)} > D_c$ **then**
10:          **[All clients → Server]** Upload $\Delta V_{i,t}$, and let $\widetilde{S}_t = \{1, 2, \cdots, N\}$
11:          **[Server]** Select incentivized participants $S_t = \mathcal{M}(\widetilde{S}_t | \widehat{D}_t)$      ▷ Incentive Mechanism
12:          **for** $i \in S_t$ **do**
13:              **[Participant $i$ → Server]** Upload $\Delta b_{i,t}$
14:              **[Server]** Update: $V_{g,t} \mathrel{+}= \Delta V_{i,t}$, $b_{g,t} \mathrel{+}= \Delta b_{i,t}$
15:                $\Delta V_{-j,t} \mathrel{+}= \Delta V_{i,t}$, $\Delta b_{-j,t} \mathrel{+}= \Delta b_{i,t}, \forall j \neq i$
16:              **[Participant $i$]** Update: $\Delta V_{i,t} = 0$, $\Delta b_{i,t} = 0$, $\Delta t_{i,t} = 0$
17:          **for** $\forall i \in [N]$ **do**
18:              **[Server → All Clients]** Download $\Delta V_{-i,t}, \Delta b_{-i,t}$
19:              **[Client $i$]** Update: $V_{i,t} \mathrel{+}= \Delta V_{-i,t}$, $b_{i,t} \mathrel{+}= \Delta b_{-i,t}$
20:              **[Server]** Update: $\Delta V_{-i,t} = 0$, $\Delta b_{-i,t} = 0$

---

Algorihtm 6 shows the incentivized communication protocol proposed by Wei et al. (2023). The arm selection strategy for client $i_t$ as time step $t$ is based on the upper confidence bound method:

$$\mathbf{x}_t = \arg\max_{\mathbf{x} \in \mathcal{A}_t} \mathbf{x}^\top \hat{\theta}_{i_t,t-1}(\lambda) + \alpha_{i_t,t-1}||\mathbf{x}||_{V_{i_t,t-1}^{-1}(\lambda)} \tag{12}$$

where $\hat{\theta}_{i_t,t-1}(\lambda) = V_{i_t,t-1}^{-1}(\lambda) b_{i_t,t-1}$ is the ridge regression estimator of $\theta_\star$ with regularization parameter $\lambda > 0$, $V_{i_t,t-1}(\lambda) = V_{i_t,t-1} + \lambda I$, and $\alpha_{i_t,t-1} = \sigma\sqrt{\log \frac{\det(V_{i_t,t-1}(\lambda))}{\det(\lambda I)} + 2\log 1/\delta} + \sqrt{\lambda}$. $V_{i_t,t}(\lambda)$ denotes the covariance matrix constructed using the data available to client $i_t$ up to time $t$.

# G  IMPLEMENTATION DETAILS

## G.1  HYPER-PARAMETER SETTINGS

As introduced in Section 3, the proposed TRUTH-FEDBAN works with any realization of the valuation function. For demonstration purpose, we instantiate it as a combination of client's weighted data collection cost plus its intrinsic preference cost, i.e., $f(\Delta V_{i,t}) = w \cdot \det(\Delta V_{i,t}) + C_i$, where $w = 10^{-4}$, and each client $i$'s intrinsic preference cost $C_i$ is uniformly sampled from $U(0, 100)$. In the simulated environment (Section 5), the time horizon is $T = 6250$, total number of clients $N = 25$, context dimension $d = 5$. We set the hyper-parameter $\epsilon = 1.0$, $\beta = 0.5$ in Algorithm 1 and Algorithm 3. The tolerance factor in Algorithm 7 is $\gamma = 1.0$.

As stated in Section 4.2, we do not assume a monopoly-free environment and thus any truthful incentive mechanism has to pay essential clients infinite incentives to guarantee their participation when necessary. Nonetheless, to visualize the impact of infinite payment, we simplify it as a constant value of $10^4$ that is orders of magnitude greater than the average participation cost. and the infinite critical value is simplified.

## G.2  CRITICAL VALUE CALCULATION FOR ALGORIHTM 3

It is not difficult to show that Algorithm 3 is also monotone and thus inherently associated with a critical payment scheme to make the resulting mechanism truthful. We now elaborate on the critical value calculation method for it. And the critical value based payment scheme for Algorithm 1 can be derived in a similar spirit.

For each client $\alpha \in S$ in the participant set $S$ (subscript $t$ is omitted for simplicity), the critical value $c_\alpha$ is determined as follows. First, rerun Algorithm 3 without client $\alpha$, i.e., setting $\widetilde{S}' = \widetilde{S} \setminus \{\alpha\}$; if the process fails to terminate with a feasible set $S'$, it suggests that client $\alpha$ is essential to satisfy the constraint, then its critical value is $c_\alpha = \infty$. Otherwise, the process can terminate and return a feasible set, denoted as $S' = \{i_1, i_2, \cdots, i_K\}$, then the critical value $c_\alpha$ is calculated by

$$c_\alpha = \max_{k \in [K]} \widehat{D}_{i_k} \cdot \frac{g(S'_{k-1} \cup \{\alpha\}) - g(S'_{k-1})}{g(S'_{k-1} \cup \{i_k\}) - g(S'_{k-1})} \tag{13}$$

where $i_k$ and $S'_k$ represent the selected client and intermediate set of $S'$ at $k$-th round. Denote $v(\alpha|S'_{k-1}) = g(S'_{k-1} \cup \{\alpha\}) - g(S'_{k-1})$, now suppose we are placing client $\alpha$ at the $k$-th position of $S'$. To do so, the maximal participation cost that client $\alpha$ can claim should satisfy that the corresponding value-to-cost ratio is higher than that of client $i_k$, i.e., $v(\alpha|S'_{k-1})/\widehat{D}_\alpha \geq v(i_k|S'_{k-1})/\widehat{D}_{i_k}$. In other words, the maximal cost client $\alpha$ can claim to replace $i_k$ is $\widehat{D}_\alpha = \widehat{D}_{i_k} \cdot v(\alpha|S'_{k-1})/v(i_k|S'_{k-1})$. Therefore, the critical value $c_\alpha$ calculated in Eq. (13) ensures that as long as the client $\alpha$ claims slightly less than $c_\alpha$, it can replace at least one client in the $K$ rounds, thus becomes selected by the server. On the contrary, if $\widehat{D}_i$ is higher than $c_\alpha$, we can show that it will by no means get selected by the server. Specifically, the condition $\hat{D}_\alpha > c_\alpha$ guarantees $\frac{g(S'_{k-1} \cup \{\alpha\}) - g(S'_{k-1})}{\hat{D}_\alpha} < \frac{g(S'_{k-1} \cup \{i_k\}) - g(S'_{k-1})}{\widehat{D}_{i_k}}, \forall k \in [K]$. We can start from the selection of the first client $k = 1$, and we want to guarantee client $\alpha$ will not be selected. The condition tells us $\frac{g(\alpha)}{\hat{D}_\alpha} < \frac{g(i_1)}{\widehat{D}_{i_1}}$, where $i_1$ denotes the client that was selected in the first place when we exclude $\alpha$. We know $\frac{g(i_1)}{\widehat{D}_{i_1}}$ is also higher than all the other clients, so algorithm will still select client $i_1$, i.e., $S_1 = \{i_1\} = S'_1$. Then for $k = 2$, the condition suggests $\frac{g(S_1 \cup \{\alpha\}) - g(S_1)}{\hat{D}_\alpha} = \frac{g(S'_1 \cup \{\alpha\}) - g(S'_1)}{\hat{D}_\alpha} < \frac{g(S'_1 \cup \{i_2\}) - g(S'_1)}{\widehat{D}_{i_2}} = \frac{g(S_1 \cup \{i_2\}) - g(S_1)}{\widehat{D}_{i_2}}$. Therefore, $\alpha$ will not be selected at $k = 2$ either, and $S_2 = S'_2$. We can show client $\alpha$ will not be selected in $S$ by induction.

## G.3  CRITICAL VALUE CALCULATION FOR ALGORIHTM 1

In contrast, there is no explicit formula to calculate the critical value in TRUTH-FEDBAN. Following Mu'Alem & Nisan (2008), we calculate the critical value using bisection search as described in Algorithm 7.

---

**Algorithm 7** Critical Value Calculation (Bisection Search)

---

**Require:** $\widetilde{S} = \{1, 2, \cdots, N\}$, $\widehat{D}_t = \{\widehat{D}_{1,t}, \widehat{D}_{2,t}, \cdots, \widehat{D}_{N,t}\}$, incentive mechanism $\mathcal{M}$, concerned
    client $i$, budget $b$, tolerance $\gamma$
1: Initialization: $L \leftarrow 0$, $H \leftarrow b$
2: **while** $\frac{H-L}{2} \geq \gamma$ **do**
3:     Calculate critical value: $c_i \leftarrow \frac{L+H}{2}$
4:     Update $\widehat{D}: \widehat{D}_{i,t} \leftarrow c_{i,t}$
5:     Run incentive mechanism: $S = \mathcal{M}(\widetilde{S}; \widehat{D})$                          ▷ Algorithm 1
6:     **if** $i \in S$ **then**
7:         $L \leftarrow c_{i,t}$
8:     **else**
9:         $H \leftarrow c_{i,t}$
10: **Return** client $i$'s critical value $c_{i,t}$

---

The idea remains the same as stated above, to calculate the critical value of a particular client, we first rerun Algorithm 1 without it in the candidate client set. If the client is essential, its critical value is $c_{i,t} = \infty$. Otherwise, we can calculate the critical value via Algorithm 7. Specifically, for any participant $i$ in the set $S_t$ found by Algorithm 1, it is clear that the bound of $i$'s critical value is its claimed cost $\widehat{D}_{i,t}$, otherwise it would not have been included in $S_t$. Denote $b$ as the terminating budget determined by Algorithm 1 when client $i$ is not considered, we can also have a upper bound for $c_{i,t} \leq b$. With the lower and upper bound as input to Algorithm 7, it has been proven (Burden et al., 2015) that the number of iterations that Algorithm 7 needs to converge to a root to within a certain tolerance $\gamma$ is bounded by $\lceil \log_2(\frac{\gamma_0}{\gamma}) \rceil$, where $\gamma_0 = |b|$.

## H    TIME COMPLEXITY ANALYSIS OF ALGORITHM 1

As the proposed Algorithm 1 includes a subroutine process of Algorithm 2, thus we start the time complexity analysis with Algorithm 2. Specifically, the worst-case time complexity of the while loop is $O(N)$. The operation inside the while loop involves finding the maximum element in a set, which takes $O(N)$ time. Therefore, the time complexity of Algorithm 2 is $O(N^2)$.

Let $M$ be the number of iterations of the while loop (Line 3) in Algorithm 1. Hence, the time complexity of Algorithm 1 is $O(M \cdot N^2)$. Specifically, the worst case is to consistently increase the budget $b$ until it reaches $\sum_{i=1}^{N} \widehat{D}_{i,t}$. Therefore, we can upper bound $M$ by considering the loop-breaking case: $b_0 \cdot (1 + \epsilon)^M \geq \sum_{i=1}^{N} \widehat{D}_{i,t}$, i.e., $M \leq \left\lceil \log_{1+\epsilon} \left( \sum_{i=1}^{N} \frac{\widehat{D}_{i,t}}{b_0} \right) \right\rceil$, where $b_0 = \min_{i \in \widetilde{S}} \widehat{D}_{i,t}$. As a result, Algorithm 1 yields the following polynomial time complexity of $O(\left\lceil \log_{1+\epsilon} \left( \sum_{i=1}^{N} \frac{\widehat{D}_{i,t}}{b_0} \right) \right\rceil \cdot N^2)$.

