# OpenReview forum: "Incentivized Truthful Communication for Federated Bandits"
_ICLR.cc/2024/Conference — ICLR 2024 poster_

### Official Review · Reviewer_Jfq6 · 2023-10-26

**Soundness:** 3 good
**Presentation:** 3 good
**Contribution:** 3 good
**Rating:** 6
**Confidence:** 2

**Summary:**

This paper extends the recent work on incentivized linear federated bandits (Wei et al. 2023) to one where the clients’ communication costs are unknown, and the server must devise a mechanism for payments that is incentive-compatible. The authors use ideas from mechanism design to design such a scheme while keeping the total payments small.

**Strengths:**

- Novel setting that combines multi-armed bandits with mechanism design
- Theoretical results seem correct and non-trivial
- I appreciate the numerical simulations that were conducted to corroborate theory

**Weaknesses:**

I find the main weakness of this paper to be the model. It is a very complex model that combines many different aspects (linear bandits, multiple clients, communication cost, incentivizing payments, truthfulness). The paper did not provide any motivating application for studying this complex model.

Regarding the motivation for the model, one of my main points of confusion is in how the reward/regret of the bandit algorithm can be compared to the incentive payment paid out by the server. Relatedly, what incentive does the server have, to pay the clients (out of pocket) to get them to participate? From my understanding, it is each client that is performing some action, and hence it is in the client’s best interest that the bandit algorithm chooses the best actions. Now, federated learning will improve the learning algorithm since more data is collected, so it is in the agent’s best interest to participate in federated learning. If the “communication cost” is higher than the benefits of participating in federated learning, then the agent can simply choose not to participate. In this paper, the server will pay such a client to participate - but what benefit does the server get when the agent participates? What if the true “communication costs” are exorbitant (e.g. $1M per communication)? The algorithm in this paper still makes the server pay, regardless of the scale of these costs.

If one is considering the problem in this paper purely for theoretical interest, the fact that the model was so complex makes it difficult to identify how the results contribute to the theory of multi-armed bandits. Most of the paper was about mechanism design, but it took a long time for me to understand the underlying federated linear bandit model - and it wasn’t clear to me which parts of the underlying bandit model were crucial and which were not.

**Questions:**

- A client’s utility, $u_{it}$ was not defined, so truthfulness (definition 1) is not well defined. Does a client’s utility involve both the regret as well as the payment? If so, how are these combined?

---

> ### Author Response · Authors · 2023-11-17
> **Response to Reviewer Jfq6 [Part 1/2]**
>
> We thank the reviewer for the constructive comments. First of all, we would like to make a fundamental clarification on the misunderstanding on the objective of **federated bandt learning**, which has caused the reviewer’s most confusions about the motivation (Q1), model behavior (Q2), and our contribution (Q3) in the context of federated bandit learning.
>
> As detailed in the general responses [CA1, CA2], in this line of research, the server’s goal is **NOT** to game or bargain with the clients. **Instead, it is the server’s desire/duty to minimize the overall regret across all clients, as achieving a near-optimal regret is in its best interest**. Imagine a distributed recommender system, where the central server’s goal is to make sure all clients’ recommendation quality is satisfactory. Using the reviewer’s language in Q2, the lower the overall regrets are, the higher the "incentives" for the server. Moreover, we note that the main research interest and challenge in “incentivized federated bandits” lies in incentive mechanism design, as explained in [CA3]. A detailed discussion of our contribution to the theory of bandit learning is presented in [CA4].
>
> Please let us know if you have any further questions, and we are more than happy to incorporate any additional suggestions that could further enhance our work and lead to a more favorable evaluation of our work.
>
>
>
> **[Q1]**: The paper did not provide any motivating application for studying this complex model.
>
> **[A1]**: The line of research on federated bandit learning has been motivated by many real-world sequential decision applications where recommender systems is a representative one. For instance, a recommendation platform (server) wants its mobile app users (clients) to opt in its new recommendation service, which switches previous on-device local bandit algorithm to a federated bandit algorithm. Although the new service is expected to improve the overall recommendation quality for all clients (minimizing the overall regret), particular clients may not be willing to participate in this collaborative learning, as the expected gain for them might not compensate their locally increased cost (e.g., communication bandwidth, added computation, lost control of their data, and etc).
>
> Therefore, a proper incentive mechanism is required to facilitate the federated learning process, as explained in [CA3]. Following this direction, our work focuses on a more realistic scenario that aims to design a truthful incentive mechanism, ensuring clients won't exploit the server’s commitment by misreporting their data sharing cost.
>
> **[Q2]**: How can the reward/regret of the bandit algorithm be compared to the incentive payment paid out by the server. Relatedly, what incentive does the server have, to pay the clients (out of pocket) to get them to participate? What benefit does the server get when the agent participates? What if the true “communication costs” are exorbitant (e.g. $1M per communication)? The algorithm in this paper still makes the server pay, regardless of the scale of these costs.
>
> **[A2]**: As explained in [A1], the server is obligated to reduce regret for all clients. **Therefore, the server’s goal is not to trade off between regret and incentive payment, but to motivate client’s participation for nearly optimal regret**. Note that this obligated server is practical in many real-world scenarios (e.g., distributed users/edge devices on a shared learning platform, different departments within the same company, etc.), where the server/platform/company's best interest is to improve the overall learning across all clients.
>
> With that being said, since Wei et al. (2023) recently opened up an interesting direction in this line of research, we absolutely agree that exploring the non-obligated server setting is a timely and important future work. The interactions between the server and clients can be modeled variously according to real-world applications. For instance, in addition to incentivizing clients for participation, the server can even charge some clients for receiving data from the server.

---

> ### Author Response · Authors · 2023-11-17
> **Response to Reviewer Jfq6 [Part 2/2]**
>
> **[Q3]**: If one is considering the problem in this paper purely for theoretical interest, the fact that the model was so complex makes it difficult to identify how the results contribute to the theory of multi-armed bandits. Most of the paper was about mechanism design, but it took a long time for me to understand the underlying federated linear bandit model - and it wasn’t clear to me which parts of the underlying bandit model were crucial and which were not.
>
> **[A3]**: Thanks for pointing out the place that could lead to unnecessary confusion. We have provided a detailed discussion on the basics objectives and challenges of federated bandits learning in [CA2].
>
> Typically, the communication scheme (i.e., arm selection strategy, communication event trigger) is a crucial part for federated bandit learning. **However, in the context of incentivized federated bandits, the main interest is on designing the incentive mechanism**, and both Wei et al (2023) [R1] and our work adopt the standard communication scheme, as we explained in [CA3].
>
> For a detailed discussion of our contribution to the theory of bandit learning, please refer to our response in [CA4].
>
> **[Q4]**: A client’s utility $u_{i,t}$,  was not defined, so truthfulness (definition 1) is not well defined. Does a client’s utility involve both the regret as well as the payment? If so, how are these combined?
>
> **[A4]**: Following Wei et al. (2023) [R1], our problem considers “individual rational” clients, meaning that client $i$ only participates in data sharing if the incentive mechanism provides a non-negative utility (see Definition 1 of Individual Rationality in [R1]), where utility is defined as the difference between incentive and true cost of client $i$ , i.e., $u_{i,t} = \mathcal I_{i,t} - D_{i,t}$, as is standard in economic analysis [R2]. We have made this notation clearer in our revised manuscript (Section 3).
>
> Nonetheless, we agree that it is also an interesting alternative to include the regret-related term as a factor in the client’s utility design. And we’d like to share some thoughts on it:
> - Under the obligated server setting, the server will help every client regardless of whether the clients share their local data or not. As the server always guarantees a nearly optimal regret in each communication round, thus there is no significant difference in such a regret-related utility term across all clients, as everyone can utilize the same amount of aggregated data shared by the server for regret reduction. Therefore, it is less meaningful to take the regret-related term in the utility design.
> - In contrast, for the non-obligated server setting, each client may receive different subsets of the aggregated data and thus obtain different levels of regret reduction. In this case, it becomes more reasonable to include the regret-related term in the utility design, which we believe is an interesting direction to explore combined with the non-obligated setting, as we discussed in [A2].
>
> **References**
>
> [R1]. Zhepei Wei, Chuanhao Li, Haifeng Xu, and Hongning Wang. Incentivized Communication for Federated Bandits. NeurIPS 2023.
>
> [R2]. Roger B Myerson and Mark A Satterthwaite. Efficient mechanisms for bilateral trading. Journal of economic theory, 29(2):265–281, 1983.

---

> > ### Comment · Reviewer_Jfq6 · 2023-11-20
> >
> > Thank you to the authors for the detailed answers. The questions I asked in my review have been clarified. I am still not convinced that this is a practical setting - in the recommendation platform example, is it actually practical that a platform pays its user for each communication round, and it pays each user differently according to their true costs? I don't mean to nitpick on this point, but I still think of this setting as mainly driven by theoretical interest. I also agree with reviewer 1gFC that the exposition of the paper is poor (which was the source of all of my questions/confusion) and hence the paper should be significantly re-written for clarity. I will maintain my score.

---

> > > ### Author Response · Authors · 2023-11-21
> > > **Response to Reviewer Jfq6 [Round 2, Part 1/2]**
> > >
> > > Thanks for the reviewer’s reply and we are glad to know the reviewer's questions have been addressed!
> > >
> > > We want to first clarify that our technical contributions have been acknowledged by all other reviewers (Reviewer 1gFC, ihns, 3EFN). For instance, Reviewer 1gFC stated that *“the setting and the contributions of the paper are indeed quite interesting (especially regarding the aspects of untruthfulness of clients and social cost, given how important these aspects are from a practical standpoint)”*, **explicitly recognizing the practical contribution of our work**. Moreover, the merit of incentivized communication under federated learning (FL) has been widely recognized by the FL community [R1, R2], and recently introduced into the bandit community [R3]. Therefore, as we explained in [Round2-A1], we believe this line of research is well motivated and of great practicality.
> > >
> > > Please let us know if you have any further questions, and we look forward to the possibility of your updated evaluation of our work.
> > >
> > > **[Round2-Q1]**: I am still not convinced that this is a practical setting - in the recommendation platform example, is it actually practical that a platform pays its user for each communication round, and it pays each user differently according to their true costs? I don't mean to nitpick on this point, but I still think of this setting as mainly driven by theoretical interest.
> > >
> > > **[Round2-A1]**: Incentivizing clients to share data or join global model training in the federated learning setting is driven by practical demands and has been extensively explored in various real-world scenarios, such as web services, edge computing, and communications [R4, R5].
> > >
> > > Take the application scenario of wireless communication [R2] for example, where the base station pays its distributed mobile users to motivate them to participate in model training, despite the fact that the base station already provides services to the mobile users. This is essentially the same situation as in our recommendation platform example, as we illustrated in the previous response [A1]. Another related practical example is the Bing search engine, where the server provides Microsoft Rewards points to incentivize its users to explore websites and generate data (clicks and query reformulations) for improved services.
> > >
> > > **Therefore, there is no doubt that this line of research is well-motivated by practical demands and has been extensively explored in the related literature and real-world applications**. Regarding the payment design, it is actually subject to the objectives behind the problem of interest. For example, if the server cannot pay too much for data collection, a reserved price strategy can be employed, i.e., any client with a critical value higher than a preset amount will not be selected by the incentive mechanism. This is a common design choice in economics [R6], which, however, if adopted in our case will compromise both the overall learning performance and the truthfulness guarantee, and is beyond the scope of our work.
> > >
> > > For more practical use cases, we refer the reviewer to the recent surveys [R4, R5].

---

> > > > ### Author Response · Authors · 2023-11-21
> > > > **Response to Reviewer Jfq6 [Round 2, Part 2/2]**
> > > >
> > > > **[Round2-Q2]**: I also agree with reviewer 1gFC that the exposition of the paper is poor (which was the source of all of my questions/confusion) and hence the paper should be significantly re-written for clarity.
> > > >
> > > > **[Round2-A2]**: As explained in our [Round2 response to reviewer 1gFC](https://openreview.net/forum?id=ykEixGIJYb&noteId=bXdGm0edoQ), in our original submission, we already explicitly introduced the necessary general background in the preliminary section (Section 3) and reminded our readers again in the method section (Section 4.2) about the important background (e.g., the communication protocol detailed in Algorithm 6), albeit leaving the details to the appendix due to the strict space limit.
> > > >
> > > > **Given that all necessary technical details and background have been included in our paper, and the reviewers found our technical contributions strong (Reviewer 1gFC, ihns, 3EFN)**, it would be very sad to find our paper got rejected, simply because such content was not presented earlier due to space limits. Moreover, we do not believe this paper has to be rewritten, as all the suggested “confusing notations” and “missing descriptions” by Reviewer 1gFC have already been explained in the original paper. Please refer to the [Round2 response to reviewer 1gFC](https://openreview.net/forum?id=ykEixGIJYb&noteId=bXdGm0edoQ) for a detailed justification.
> > > >
> > > > With that being said, we have included the table in [CA4] in our latest manuscript so that readers with different backgrounds can readily grasp our technical contribution with minimal effort.
> > > >
> > > > While we understand that the reviewer may have had some confusions regarding our paper, we would appreciate specific details on which part of our current exposition is still unclear. Please let us know, and we are more than happy to provide additional clarifications that could further enhance our work and lead to a more favorable evaluation.
> > > >
> > > > **References**
> > > >
> > > > [R1]. Karimireddy, Sai Praneeth, Wenshuo Guo, and Michael I. Jordan. Mechanisms that incentivize data sharing in federated learning. arXiv preprint arXiv:2207.04557 (2022).
> > > >
> > > > [R2]. T. H. Thi Le et al. An Incentive Mechanism for Federated Learning in Wireless Cellular Networks: An Auction Approach. IEEE Transactions on Wireless Communications, 2021.
> > > >
> > > > [R3]. Zhepei Wei, Chuanhao Li, Haifeng Xu, and Hongning Wang. Incentivized Communication for Federated Bandits. NeurIPS 2023.
> > > >
> > > > [R4]. Jian Pei. A survey on data pricing: from economics to data science. IEEE Transactions on knowledge and Data Engineering, 34(10):4586–4608, 2020.
> > > >
> > > > [R5]. Xuezhen Tu, Kun Zhu, Nguyen Cong Luong, Dusit Niyato, Yang Zhang, and Juan Li. Incentive mechanisms for federated learning: From economic and game theoretic perspective. IEEE Transactions on Cognitive Communications and Networking, 2022.
> > > >
> > > > [R6]. Ostrovsky, Michael, and Michael Schwarz. Reserve prices in internet advertising auctions: A field experiment. In Proceedings of the 12th ACM conference on Electronic commerce, pp. 59-60. 2011.

---

> > > > > ### Comment · Reviewer_Jfq6 · 2023-11-22
> > > > >
> > > > > Thank you to the authors again for the detailed responses. Though I still feel the exposition can be improved, I acknowledge the space constraints as well as the existing literature on federated bandits. I have increased my score.

---

> > > > > > ### Author Response · Authors · 2023-11-22
> > > > > >
> > > > > > We would like to express our sincere gratitude to the reviewer for his/her active engagement in the entire review process. Meanwhile, we are also very glad to find our paper is now positively evaluated by the reviewer, which would give us the invaluable opportunity to share our encouraging research findings with the community!
> > > > > >
> > > > > > We acknowledge there is still room for improvement in the exposition of our paper due to the strict page limit and our choice in balancing between the discussion of background and our technical contributions. And we are committed to improving it based on the valuable suggestions from our reviewers. To help our readers more easily comprehend and appreciate our work, as suggested by Reviewer Jfq6, we will add more justifications about the problem setting of incentivized federated bandit learning, especially from practical perspectives, as well as include more background details in the paper’s main body.

---

### Official Review · Reviewer_3EFN · 2023-10-29

**Soundness:** 3 good
**Presentation:** 2 fair
**Contribution:** 3 good
**Rating:** 8
**Confidence:** 4

**Summary:**

The work builds on, as an extension of Wei et al. (2023), to incentivize the truthful participation of clients to improve overall utility for each client. The authors show the developed communication protocol TRUTH-FEDBAN enjoys near-optimal theoretical guarantees on regret and communication costs.

**Strengths:**

This work presents a mechanism design to ensure the truthful participation of clients, where the client's only beneficial strategy is to share their true costs in a federated bandit learning setting. The incentive-compatible communication protocol offers near-optimal theoretical guarantees on regret and communication costs. Numerical evaluations validate the method's efficacy.

**Weaknesses:**

1. I am a bit unsure of the contribution and the motivation for the developed solution of this work, particularly considering how this work is built (as an extension) on Wei et al. (2023); hence, the claim for making this work a first attempt in mechanism design for federated bandit learning seems incorrect? Also, for the method developed, several recent works on federated settings, such as FL, tackle such issues for truthful and fair participation, employing economic tools. Contract Theory-based methods or Auction-based methods have been "extensively" employed in mechanism design for truthful interaction between the clients and the server. For instance, [R1, R2].

[R1] Karimireddy, Sai Praneeth, Wenshuo Guo, and Michael I. Jordan. "Mechanisms that incentivize data sharing in federated learning." arXiv preprint arXiv:2207.04557 (2022).
[R1] T. H. Thi Le et al., "An Incentive Mechanism for Federated Learning in Wireless Cellular Networks: An Auction Approach," in IEEE Transactions on Wireless Communications, vol. 20, no. 8, pp. 4874-4887, Aug. 2021, doi: 10.1109/TWC.2021.3062708

In that reference, I am not sure relaxing the data sharing cost with a valuation function, commonly used in standard economic analysis, is a sufficient contribution to this work. Can you explain more about the missing guarantees on the social cost of Wei et al. (2023)? Later, after Def. 2, you mentioned the definition of social cost in this work is different than theirs.

2. Following my earlier comment, the setup and the interaction procedure is unclear to establish the contribution, again, as compared to Wei et. al., 2023. The preliminary model is not rigorous to that end. For instance, what exactly is pulling the arm characterised in Section 3 in a federated bandit setting? How to interpret y_t following it? what is w in footnote 1? and so on. This can be significantly improved.

3. Can the authors support the claim that "more frequent communication leads to lower regret"? (and the line that follows in pg. 3) It is understandable in terms of the communication overhead, but this is also the fact that you gain in training efficiency. This leads back to my earlier confusion regarding how "regret" is quantified.

4. Is the critical value defined in Def. 4 unique?
5. I must admit the proof of monotonicity has not been conveyed clearly in its current form; can you provide a discussion on this?
6. Simulations:
- The methods build on Wei et. al. 2023 but didn't use it as a baseline.
- What about the time-complexity analysis and the overall learning performance following the proposed approach? Also, how well the method scales.
- The general setup with "sequential interaction" is limiting in itself, in my understanding.

**Questions:**

Please consider the questions raised in the weakness section.

---

> ### Author Response · Authors · 2023-11-17
> **Response to Reviewer 3EFN [Part 1/3]**
>
> We thank the reviewer for the constructive comments. **Unfortunately, it appears that there are some critical confusions regarding our studied problem (e.g., as in [Q2, Q5, Q10]), thus leading to a misunderstanding of our contribution (e.g., as in [Q1, Q4])**. We should first clarify that our focus is on incentive mechanism design for **federated bandit learning**, where “sequential interaction” features the core nature of this online learning problem, as opposed to the general federated learning that operates on a fixed offline dataset. Please refer to our general response [CA1] for detailed clarification on this. Moreover, the standard communication scheme (discussed in [CA3]) used in our work is **NOT** our claimed contribution. Please see a detailed discussion on our contribution in [CA4].
>
> Please let us know if you have any further questions, and we are more than happy to incorporate any additional suggestions that could further enhance our work and lead to a more favorable evaluation of our work.
>
> **[Q1]**: I am a bit unsure of the contribution and the motivation for the developed solution of this work, particularly considering how this work is built (as an extension) on Wei et al. (2023); hence, the claim for making this work a first attempt in mechanism design for federated bandit learning seems incorrect?
>
> **[A1]**: Thanks for pointing out the place that could lead to unnecessary confusions. The truthfulness guarantee is a crucial property in mechanism design, especially in the context of **federated bandit learning** (see definition in [CA1]). Motivated by this practical demand, our work provides the first solution that simultaneously ensures both truthfulness and nearly optimal learning performance. Please also refer to the motivating example in our response to [Q1 of Reviewer Jfq6].
>
> We acknowledge that Wei et al. (2023) [R6] is the first work to investigate mechanism design in the incentivized federated bandits. Indeed, in our introduction section, we only claim to present "**the first incentive-compatible (i.e., truthful)** communication protocol", signifying that our work is the first solution to achieve nearly optimal performance without relying on the truthfulness assumption in the incentivized federated bandit problem.
>
> Please refer to our general responses for a detailed comparison on the commonality (i.e., using the same communication scheme, see [CA3]) and differences (i.e., our unique contribution, see [CA4]).
>
> **[Q2]**: Also, for the method developed, several recent works on federated settings, such as FL, tackle such issues for truthful and fair participation, employing economic tools. Contract Theory-based methods or Auction-based methods have been "extensively" employed in mechanism design for truthful interaction between the clients and the server. For instance, [R1, R2]. In that reference, I am not sure relaxing the data sharing cost with a valuation function, commonly used in standard economic analysis, is a sufficient contribution to this work.
>
> **[A2]**: Thanks for bringing up the related works. Indeed, there has been growing research effort in exploring incentivized data sharing protocols in federated learning [R3, R4]. However, most of them, including [R1, R2], only focused on the supervised offline learning setting. **As we discussed in [CA1], none of these solutions can be applied to the federated bandit learning problem**. Wei et al. (2023) [R6] is the only notable exception, but however, it still relies on the unrealistic truthfulness assumption as we explained above. For a detailed discussion on the contributions of our work, please refer to our response [CA4].
>
> Regarding the relaxation on data sharing cost $D_{i,t} = f(\Delta V_{i,t})$, we would like to clarify that the intended purpose is to generalize our solution to any form of data sharing cost, rather than restricting it to a constant value as in Wei et al. (2023) [R6]. In fact, none of our analysis relies on the specific form of $D_{i,t} = f(\Delta V_{i,t})$ - all that matters is that client $i$ has  value $D_{i,t}$ for its data at time step $t$, and we do not impose any conditions on the function $f$. Please see also our response to [Q3 of Reviewer 1gFC] for further clarification on this aspect.

---

> ### Author Response · Authors · 2023-11-17
> **Response to Reviewer 3EFN [Part 2/3]**
>
> **[Q3]**:  Can you explain more about the missing guarantees on the social cost of Wei et al. (2023)? Later, after Def. 2, you mentioned the definition of social cost in this work is different than theirs.
>
> **[A3]**: In the remarks following Definition 2, we stated that the social cost defined in our work is different from the incentive cost studied in Wei et al. (2023) [R6]. Specifically,
> - **Social Cost**: the total actual costs incurred by all participating clients in the incentivized client set $S_t$, i.e., $\sum_{i \in S_t} D_{i,t}$.
> - **Incentive Cost**: the total payment (monetary incentive) the server made to all participating clients in the incentivized client set $S_t$, i.e., $\sum_{i \in S_t} \mathcal I_{i,t}$.
>
> As truthfulness is assumed in their setting, the server simply pays the participating client’s claimed cost, i.e., $\mathcal I_{i,t} = D_{i,t}$. In contrast, we do not assume truthfulness in our setting. To ensure truthfulness, the server needs to pay the calculated critical value to the participating client, which is guaranteed to be no less than their true cost (see Theorem 8), i.e., $\mathcal I_{i,t} \geq D_{i,t}$. Therefore, in our setting, these two costs are completely different concepts.
>
> Regarding the missing guarantee on the social cost in Wei et al. (2023) [R6], even under the truthfulness assumption, their proposed incentive mechanism (Algorithm 3 in their paper) only satisfies the constraint of the optimization problem defined in Eq. (1), while providing no guarantee on the objective. **In other words, their method aims solely to find a feasible set of clients to collect data so as to achieve near-optimal regret, without ensuring the optimality of cost from the selected set** -  ideally, the optimal set should satisfy the constraint while yielding the lowest social cost.
>
> **[Q4]**: Following my earlier comment, the setup and the interaction procedure is unclear to establish the contribution, again, as compared to Wei et. al., 2023. The preliminary model is not rigorous to that end. For instance, what exactly is pulling the arm characterised in Section 3 in a federated bandit setting? How to interpret y_t following it? what is w in footnote 1?
>
> **[A4]**: In the context of federated bandits (introduced in Section 3 of our paper), there are $N$ clients interacting with the environment to learn the unknown parameter, coordinated by a central server. At each time step $t$, an arbitrary client becomes active to pull an arm $x_t$, and receives the corresponding reward $y_t$ from the environment. A vivid illustration of the interaction procedure can be found in Figure 1(a) of [R5].
>
> As is standard in federated bandit learning, we use the upper confidence bound (UCB)-based arm selection strategy to guide the client’s interaction with the environment, where the collected data $(x_t, y_t)$ is used for model estimation, as detailed in [CA2]. Regarding the communication scheme between the server and clients under the incentivized setting, please see our response [CA3].
>
> The notation $w$ in footnote 1 represents how much client $i$ weighs its data collection cost $\det(\Delta V_{i,t})$ in its data-sharing cost function $D_{i,t} = f(\Delta V_{i,t})$. However, as we clarified in [A2] and in our response to Q3 of Reviewer 1gFC, the specific form is trivial and will be removed from our final version to avoid further confusion.
>
> **[Q5]**: Can the authors support the claim that "more frequent communication leads to lower regret"? (and the line that follows in pg. 3). It is understandable in terms of the communication overhead, but this is also the fact that you gain in training efficiency. This leads back to my earlier confusion regarding how "regret" is quantified.
>
> **[A5]**: As in line with the established common ground and convention of this line of research, our claim that "more frequent communication leads to lower regret" and the definition of “regret” are the standard consensus/terminology shared by federated bandits literature. Please refer to our response [CA2] for a detailed clarification.
>
> **[Q6]**: Is the critical value defined in Def. 4 unique?
>
> **[A6]**: Yes. Suppose $c_{i,t}^{1}$ and $c_{i,t}^{2}$ are two distinct critical values for client $i$ at time step $t$. When client $i$ claims its data sharing cost as $\widehat D_{i,t} = (c_{i,t}^{1} + c_{i,t}^{2}) / 2$, then client $i \in S_t$ and $i \notin S_t$ will simultaneously hold true according to Definition 4, which is clearly a contradiction. Therefore, by Definition 4, the critical value for each client $i$ is unique.

---

> ### Author Response · Authors · 2023-11-17
> **Response to Reviewer 3EFN [Part 3/3]**
>
> **[Q7]**: I must admit the proof of monotonicity has not been conveyed clearly in its current form; can you provide a discussion on this?
>
> **[A7]**: As defined in Definition 3, Algorithm 1 is monotone if for any selected participating client $i$ with a reported data sharing cost $\widehat D_{i,t}$, it will remain selected by Algorithm 1 whenever it reports a lower value, i.e., $\widehat D^\prime_{i,t} \leq \widehat D_{i,t}$.
>
> Note that Algorithm 1 searches for the participating set $S_t$ by starting from a minimal budget (Line 2) and repeatedly increases the budget (Line 4) until the resulting client set found by Algorithm 2 satisfies the specified condition (Line 3). Therefore, decreasing a client's reported cost $\widehat D_{i,t} \rightarrow \widehat D^\prime_{i,t}$ can cause a different output by Algorithm 2 and, consequently, terminate the search process of Algorithm 1 at a different budget $b \rightarrow b'$ with a potentially different selection of the participant set ${S_t} \rightarrow {S^\prime_t}$.
>
> **As a result, to prove monotonicity, we need to show that given $i \in S_t$ when client $i$ reports $\widehat D_{i,t}$, the proposition $i \in S^\prime_t$ always holds true when $\widehat D^\prime_{i,t} \leq \widehat D_{i,t}$**. This is done by demonstrating that the resulting objective value $g_t({S_t})$ from Algorithm 2 is non-decreasing with respect to its input budget $b$, as we proved in Lemma 12. In other words, it becomes impossible for $b' > b$ because otherwise the search process in Algorithm 1 would have stopped at $b$. Since decreasing client $i$'s claimed cost will increase its chance of getting selected by Algorithm 2 (Line 3), we can show that $i \in S'_t$ always holds true for all $b' \leq b$, as explained in Appendix A. This finishes the proof.
>
> **[Q8]**: The methods build on Wei et. al. 2023 but didn't use it as a baseline in simulation
>
> **[A8]**: We have to clarify that despite achieving the same results on regret and communication cost, our incentive mechanism is completely different from that of Wei et al. (2023) [R6], and **these two are not comparable as their method builds on the truthfulness assumption, which is absent in our setting**. Please find a detailed comparison in our general response [CA4].
>
> **[Q9]**: What about the time-complexity analysis and the overall learning performance following the proposed approach? Also, how well the method scales.
>
> **[A9]**: As detailed in the response to [Q4 of Reviewer 1gFC], the proposed method holds a polynomial time complexity of $O\left(\left\lceil \log_{1+\epsilon}\left(\sum_{i=1}^{N} \frac{\widehat{D}{i,t}}{b_0}\right)\right \rceil \cdot N^2\right)$, where $b_0 = \min_{i \in \widetilde S} \widehat D_{i,t}$, scaling quadratically with the number of clients $N$.
>
> Our method archives nearly-optimal overall learning performance with $O(d\sqrt{T} \log T)$ regret and $O(N^{2} d^3\log T)$ communication cost, which is confirmed by our empirical study in Section 5. For example, as Figure 1(a) shows, the regret curve of our proposed method converges after ~1000 iterations, indicating that the algorithm already learned an accurate estimation of the unknown parameter and can consistently choose the best arm, thus incurring no regret.
>
> **[Q10]**: The general setup with "sequential interaction" is limiting in itself, in my understanding.
>
> **[A10]** : We should clarify that our studied problem is **federated bandit learning**, which is an important research field with a wide range of real-world applications featuring sequential decision-making processes, as discussed in [CA1]. Therefore, investigating the incentive design with this setup holds significant value both in research and practical applications.
>
> **References**
>
>
> [R1]. Karimireddy, Sai Praneeth, Wenshuo Guo, and Michael I. Jordan. "Mechanisms that incentivize data sharing in federated learning." arXiv preprint arXiv:2207.04557 (2022).
>
> [R2]. T. H. Thi Le et al., "An Incentive Mechanism for Federated Learning in Wireless Cellular Networks: An Auction Approach," in IEEE Transactions on Wireless Communications, vol. 20, no. 8, pp. 4874-4887, Aug. 2021, doi: 10.1109/TWC.2021.3062708
>
> [R3]. Jian Pei. A survey on data pricing: from economics to data science. IEEE Transactions on knowledge and Data Engineering, 34(10):4586–4608, 2020.
>
> [R4]. Xuezhen Tu, Kun Zhu, Nguyen Cong Luong, Dusit Niyato, Yang Zhang, and Juan Li. Incentive mechanisms for federated learning: From economic and game theoretic perspective. IEEE Transactions on Cognitive Communications and Networking, 2022.
>
> [R5]. Chuanhao Li, and Hongning Wang. Asynchronous Upper Confidence Bound Algorithms for Federated Linear Bandits. AISTATS 2022.
>
> [R6]. Zhepei Wei, Chuanhao Li, Haifeng Xu, and Hongning Wang. Incentivized Communication for Federated Bandits. NeurIPS 2023.

---

> > ### Comment · Reviewer_3EFN · 2023-11-19
> >
> > Thanks to the authors for a detailed response. I agree with some of the concerns raised by, particularly Reviewer 1gFC, and others regarding the contribution of this work compared to Wei et al. (2023) while acknowledging the distinction made by authors in [CA4]. It might be correct, the claim being the first one to simultaneously achieve IC and optimal regret in federated bandit learning (pls. note I am aware of the distinction between federated learning and federated bandit learning), but the authors should discuss several existing methods employing contract-theory/auction mechanism to enforce truthful participation in such a setting (collaborative learning/FL) to position their arguments better. This can be improved in my understanding. Further, following your responses,
> > - the authors mentioned their method can generalize for any form of valuation function $f(\cdot)$. Can the authors justify this claim?
> > - you mentioned "all the client's secret lies in $\Delta b_{i,t}$" and revealing $\Delta V_{i,t}$ won't compromise the client's "data privacy". I believe this needs further discussion/clarification.
> > - and I wonder, isn't this one of the limiting criteria that all clients need to participate and report $\Delta V_{i,t}$?

---

> > > ### Author Response · Authors · 2023-11-20
> > > **Response to Reviewer 3EFN [Round 2, Part 1/3]**
> > >
> > > Thank you for the further clarification and the constructive suggestions! We have included new discussion on the related works employing contract-theory and auction mechanisms to enforce truthful participation under the federated learning setting. Please refer to our response [Round2-A1] and Section 2 of our latest manuscript for a detailed discussion.
> > >
> > > **[Round2-Q1]**: The authors should discuss several existing methods employing contract-theory/auction mechanism to enforce truthful participation in such a setting (collaborative learning/FL) to position their arguments better.
> > >
> > > **[Round2-A1]**: Thanks for the suggestion to enhance our positioning and contributions. As suggested by the reviewer, there have been growing efforts on investigating truthful incentive mechanisms in the context of *federated learning*.
> > >
> > > - One notable recent work is Karimireddy et al. (2022) [R1], where a contract-theory based incentive mechanism is introduced to maximize data sharing while avoiding free-riding clients who contribute no data but still enjoy an improved model. Specifically, instead of paying monetary incentives, the server uses the global model trained on aggregated data as incentive to motivate clients sharing their data, in which the more data one client contributes the better model it will receive from the server. More specifically, every client gets different snapshots of the global model with different levels of accuracy, and truthfully reporting their data sharing costs is their best response under the proposed incentive mechanism (namely, “accuracy-shaping”). As a result, there is no overall performance guarantee in this mechanism and their focus is on investigating the level of accuracy the system can achieve (i.e., maximizing the total amount of shared data) under this truthful incentive mechanism without additional payment.
> > >
> > > In contrast, as we clarified in [CA2] and [A1 to Reviewer Jfq6], in our problem setting **the server is obligated to improve the overall performance of the bandit learning system**, i.e., obtaining nearly optimal regret for all clients. To achieve this goal, the server chooses to pay a subset of clients to collect data with respect to an information constraint and minimal social cost objective, forming the optimization problem Eq.(1).
> > >
> > > - A closely related work to ours is Le et al. (2021) [R2] that investigates truthful mechanism design in the application scenario of wireless communication, where they focus on the uplink bandwidth allocation problem between the base station (server) and distributed mobile users (clients). Specifically, as bandwidth is limited, the server needs to carefully allocate the resources among the clients, i.e., the server pays a subset of clients to upload their local models, which is formulated as an auction game in their work. Specifically, their server’s goal is to maximize the system’s social welfare, which is defined as the difference between the server’s satisfaction level and participating clients’ claimed costs, with respect to a knapsack upper bound constraint, as defined in Eq.(30) of their paper.
> > >
> > > However, our server aims at minimizing the social cost, which is defined as the actual costs among all participating clients, with respect to a submodular lower bound constraint, as defined in Eq.(3) of our manuscript. The fundamental difference behind the two servers’ goals is that their problem formulation places the overall federated learning performance (referred to as 'global accuracy') as the optimization objective, with a limited amount of resources to be allocated to clients (clients use the resources to train local models for better local accuracy to share with the server) as the constraint. As a result, they do not have a guarantee on the overall learning performance.
> > >
> > > In contrast, we formulate the optimization problem where the overall federated learning performance (i.e., the determinant ratio in Eq. (3)) serves as a constraint, and the objective is to minimize the social cost. In this way, we can guarantee the overall federated learning performance, which is indispensable for achieving near-optimal regret in our work as we explained in Section 3. These different purposes lead to different requirements on mechanism design. Therefore, despite the fact that we share a similar idea of using a monotone participant selection rule and critical-value based payment design to guarantee truthful cost reporting, **the underlying fundamental optimization problems are completely different, and consequently their solution cannot be used to solve our problem**.
> > >
> > > Besides pursuing the truthfulness guarantee in incentive mechanism design under the collaborative/federated learning setting, the other related line of research focuses on designing incentive mechanisms that ensures fairness among collaborative clients [R3, R4, R5, R6], which is also an important direction, despite being beyond the scope of our work.

---

> > > > ### Author Response · Authors · 2023-11-20
> > > > **Response to Reviewer 3EFN [Round 2, Part 2/3]**
> > > >
> > > > **[Round2-Q2]**: The authors mentioned their method can generalize for any form of valuation function $f(\cdot)$. Can the authors justify this claim?
> > > >
> > > > **[Round2-A2]**:
> > > > The client’s participation cost stems from multiple intrinsic aspects such as the client’s individual communication resource consumption, data production cost, or potential privacy loss, and we follow the terminology in Wei et al. (2023) to refer to it as the "data sharing cost".
> > > >
> > > > In essence, the data sharing cost is a general design that characterizes the “unwillingness” of clients in data sharing, and is represented by a scalar $D_{i,t}$. Though in our work, we do not assume how clients determine their intrinsic cost, the previously mentioned notation of $D_{i,t} = f(\Delta V_{i,t}) = w \cdot \det(\Delta V_{i,t}) + C_i$ may leave the readers with a wrong impression that our method adheres to this specific form. As noted in the previous response [A2], we will remove this notation in the final manuscript.
> > > >
> > > > More importantly, our proof (e.g., Proposition 5, Theorem 8, Theorem 9, Theorem 10) works with any arbitrary value of $D_{i,t}$, regardless of the function form of $f(\cdot)$. **Therefore, any form of the valuation function $f(\cdot)$ can be applied to our method, and none of our results or guarantees will be affected by a different design of $f(\cdot)$**.

---

> > > > > ### Author Response · Authors · 2023-11-20
> > > > > **Response to Reviewer 3EFN [Round 2, Part 3/3]**
> > > > >
> > > > > **[Round2-Q3]**: You mentioned "all the client's secret lies in $\Delta b_{i,t}$ and revealing $\Delta V_{i,t}$ won't compromise the client's "data privacy". I believe this needs further discussion/clarification. Isn't this one of the limiting criteria that all clients need to participate and report $\Delta V_{i,t}$?
> > > > >
> > > > > **[Round2-A3]**: As we explained in [CA2], in federated bandit learning, to construct an estimator $\hat \theta_{t}$ of the unknown model parameter $\theta_{\star}$, clients should use ridge regression $\hat \theta_{t} = V_{t}^{-1} b_{t}$, where $V_{t}=\sum_{s=1}^{t} \mathbf x_{s} \mathbf x_{s}^{\top}$, and $b_{t}=\sum_{s=1}^{t} \mathbf x_{s} y_{s}$, which necessitate eliciting both $V_{i,t}$ and $b_{i,t}$ from the clients to construct the global estimation.
> > > > >
> > > > > As detailed in [CA3], all clients first upload their $\Delta V_{i_t,t}$ before the execution of the incentive mechanism, and only participating clients (those who are properly incentivized) will upload their $\Delta b_{i,t}$ to the server. Note that the clients’ collected data $(\mathbf x_{t}, y_{t})$ are always stored locally and never shared with the server. This design enables the server to calculate the corresponding incentive based on part of the client’s local updates (i.e., $\Delta V_{i_t,t}$) without direct access of the client’s data, as in line with Wei et al. (2023) [R9].
> > > > >
> > > > > The claim “all the client's secret lies in $\Delta b_{i,t}$ and revealing $\Delta V_{i,t}$ won't compromise the client's data privacy" directly follows the fact that only having $\Delta V_{i,t}$ is not sufficient to construct the estimator or recover any client’s local data, which can only be done together with the access of $b_{i,t}$ that contains the information of reward $y_t$.
> > > > >
> > > > > **Therefore, the clients’ revelation of $\Delta V_{i,t}$ before their commitment to data sharing essentially brings no harm to their data privacy and is a standard design in incentivized federated bandit learning to facilitate incentive calculation [R9]**. Even beyond the context of incentivized federated bandit learning, this revelation of $\Delta V_{i,t}$ is a well-established convention as a means to control communication in federated linear bandits [R7, R8], and is not considered as limiting criteria in this line of research.
> > > > >
> > > > > Please let us know if you have any further questions, and we look forward to the possibility of your updated evaluation of our work.
> > > > >
> > > > > **References**
> > > > >
> > > > > [R1]. Karimireddy, Sai Praneeth, Wenshuo Guo, and Michael I. Jordan. Mechanisms that incentivize data sharing in federated learning. arXiv preprint arXiv:2207.04557 (2022).
> > > > >
> > > > > [R2]. T. H. Thi Le et al. An Incentive Mechanism for Federated Learning in Wireless Cellular Networks: An Auction Approach. IEEE Transactions on Wireless Communications, 2021.
> > > > >
> > > > > [R3]. Avrim Blum, Nika Haghtalab, Richard Lanas Phillips, Han Shao.  One for One, or All for All: Equilibria and Optimality of Collaboration in Federated Learning. ICML 2021.
> > > > >
> > > > > [R4]. Xinyi Xu, Lingjuan Lyu, Xingjun Ma, Chenglin Miao, Chuan Sheng Foo, and Bryan Kian Hsiang Low. Gradient driven rewards to guarantee fairness in collaborative machine learning. NeurIPS 2021.
> > > > >
> > > > > [R5]. Rachael Hwee Ling Sim, Yehong Zhang, Mun Choon Chan, and Bryan Kian Hsiang Low. Collaborative machine learning with incentive-aware model rewards. ICML 2020.
> > > > >
> > > > > [R6]. Kate Donahue and Jon Kleinberg. Fairness in model-sharing games. WWW 2023.
> > > > >
> > > > > [R7]. Yuanhao Wang, Jiachen Hu, Xiaoyu Chen, and Liwei Wang. Distributed bandit learning: Near-optimal regret with efficient communication. ICLR 2020.
> > > > >
> > > > > [R8]. Chuanhao Li and Hongning Wang. Asynchronous upper confidence bound algorithms for federated linear bandits. AISTATS 2022. ​​
> > > > >
> > > > > [R9]. Zhepei Wei, Chuanhao Li, Haifeng Xu, and Hongning Wang. Incentivized Communication for Federated Bandits. NeurIPS 2023.

---

> > > > > > ### Comment · Reviewer_3EFN · 2023-11-22
> > > > > >
> > > > > > I thank the authors for their diligent efforts in clarifying potential confusion during the rebuttal period. The topic is certainly of great interest. While I have some reservations about the organization of the work (improvements required in positioning the problem, the interaction setting, and their contributions - as raised by other reviewers as well), most of my comments have been addressed. Following your revisions (incl. Table 1) and responses to the other reviewer's comments, I think the merit and contributions of this work stand out well; thus, I am raising my scores. All the best!

---

> > > > > > > ### Author Response · Authors · 2023-11-22
> > > > > > >
> > > > > > > We highly appreciate and respect the reviewer’s devotion and active engagement in the entire review process! It is encouraging to find that the merits and contributions of our work are well comprehended and highly appreciated by the reviewer! In fact, we are very excited about our results and findings in this work, and we truly look forward to its contributions to the advancement of federated bandit learning.
> > > > > > >
> > > > > > > In the meantime, we deeply appreciate and agree with the reviewer’s suggestions to include more discussions on related works for better positioning of the investigated problem, and to incorporate more background details into the paper’s main body to help readers easily comprehend our work and appreciate our unique scientific contributions. We will carefully improve the paper’s organization, making it more accessible for broader groups of audiences to comprehend and appreciate our research.

---

### Official Review · Reviewer_ihns · 2023-11-01

**Soundness:** 3 good
**Presentation:** 3 good
**Contribution:** 3 good
**Rating:** 6
**Confidence:** 2

**Summary:**

The paper studies a federated learning problem with N strategic, individually rational agents that repeatedly interact with an environment over T rounds to receive rewards. Every agent faces the same environment characterized by a common latent parameter and stochastic rewards that are linear in the action with additive zero mean sub-Gaussian noise. Each agent wants to minimize her regret over the T rounds of interaction, subject to communication costs. The authors propose a truthful mechanism that incentivizes agents via payments (by a central server) to communicate and report their true participation costs, and simultaneously guarantees $\tilde{\mathcal{O}}\left( \sqrt{T} \right)$ individual regret.

**Strengths:**

The main contribution, in my assessment, is an improvement over Wei et al. (2023) in proposing a truthful incentive mechanism (for reporting participation costs) that simultaneously guarantees $\tilde{\mathcal{O}}\left( \sqrt{T} \right)$ near-optimal learning loss.

**Weaknesses:**

Given that this work is based off of the model of Wei et al. (2023) who essentially formulate all of the problem setting and the optimization problem being solved, the contributions could run the risk of being seen as incremental.

However, I believe this paper provides an improved and richer solution concept in comparison to that proposed in Wei et al. (2023) by factoring in regret, social cost and as well as the incentives involved in reporting participation costs.

**Questions:**

Is the bi-criteria approximation in Theorem 9 best possible or it can possibly be improved to $[1+\epsilon, 1-\delta(\epsilon)]$, where $\delta(\epsilon)$ decreases in $\epsilon$ with $\delta(\infty) = 0$. Also, is $\delta(0) = \frac{1}{e}$ best possible?

---

> ### Author Response · Authors · 2023-11-17
> **Response to Reviewer ihns**
>
> We thank the reviewer for the appreciation of our work. Please let us know if you have any further questions, and we are more than happy to incorporate any additional suggestions that could further enhance our work and lead to a more favorable evaluation of our work.
>
> **[Q1]**: The contributions of this work could run the risk of being seen as incremental to Wei et al. (2023). However, I believe this paper provides an improved and richer solution concept in comparison to that proposed in Wei et al. (2023) by factoring in regret, social cost and as well as the incentives involved in reporting participation costs.
>
> **[A1]**: Despite achieving the same results on regret and communication cost as Wei et al. (2023), our solution is completely different from theirs, with fewer assumptions yet richer property guarantees. Please see a detailed discussion on our contribution in [CA4].
>
>
> **[Q2]**: Is the bi-criteria approximation in Theorem 9 best possible or it can possibly be improved to $[1+\epsilon, 1-\delta(\epsilon)]$ , where $\delta(\epsilon)$ decreases in $\epsilon$ with $\delta(\infty) = 0$. Also, is $\delta(0) = \frac{1}{e}$ best possible?
>
> **[A2]**: The approximation factor of $1-e^{-1}$ on the constraint in Theorem 9 stems from the approximation factor of the solution to the sub-problem defined in Line 5 of Algorithm 1, which is an NP-hard submodular optimization problem. In our method, we resort to the greedy algorithm (Algorithm 2) that holds the approximation factor of $1-e^{-1}$. This performance guarantee is known as best possible for this sub-problem [R1].
>
> Regarding the objective approximation factor, our result of $(1+\epsilon)$ is the best possible in terms of order-wise analysis, as the positive constant $\epsilon > 0$ can be arbitrarily small. For constant-wise analysis, this is still an open question in the field of submodular optimization to the best of our knowledge. With that being said, **any possible constant-wise change in our bi-criterion approximation will not impact our upper bound analysis or the truthfulness guarantee of our proposed method**.
>
> **Reference**
>
> [R1]. Maxim Sviridenko. A note on maximizing a submodular set function subject to a knapsack constraint. Operations Research Letters, 32(1):41–43, 2004.

---

> > ### Comment · Reviewer_ihns · 2023-11-20
> > **Keeping my score**
> >
> > I thank the authors for answering my queries and for providing a very comprehensive overall rebuttal. Upon examining the points that were raised, I do concur that the paper could substantially benefit from being rewritten in a way that precisely distills out contributions vis-a-vis Wei et al. (2023). Personally, I found this paper hard to review as I was not aware of Wei et al. (2023) before, and current exposition implicitly assumes reader's familiarity with prior work. That said, I'd imagine this wouldn't be an issue for many as well. I'll therefore maintain my score.

---

### Official Review · Reviewer_1gFC · 2023-11-08

**Soundness:** 3 good
**Presentation:** 2 fair
**Contribution:** 3 good
**Rating:** 8
**Confidence:** 4

**Summary:**

The paper studies the problem of learning the unknown parameter in linear parametric bandits via federated/distributed learning (FL) with a central server and multiple clients, when each client must be incentivised to participate in the learning task. More specifically, each client possesses an intrinsic participation cost (e.g., amount of local computational resources at the client required to execute its share of the FL task), and participates only when its incentive for participation exceeds its participation cost. The paper studies the interesting and practically relevant setting when each client may potentially misreport its participation cost (in the interest of exploiting the system or maximising its utility), a setting that is not studied in the prior works. The authors borrow the ideas of *truthfulness* and *social cost* (popular in economics) to design an FL algorithm in which each client can maximise its utility only if it reports the true participating cost. For such an algorithm, the authors obtain bounds on the communication cost and pseudo-regret for FL with incentivised communication, while also demonstrating that the optimal social cost may be achieved up to scaling factors.

**Strengths:**

The paper studies the interesting and practically relevant setting when each client may potentially misreport its participation cost (in the interest of exploiting the system or maximising its utility), a setting that is not studied in the prior works. For the problem of FL with incentivisation studied in the prior work of Wei et al. (2023), the paper brings in the ideas of *truthfulness* and *social cost* from economics to quantify the performance of a regret-minimisation algorithm. A formal demonstration of the monotonicity of Algorithm 1 is one of the key contributions of the paper; a similar result is missing in the prior works on FL. The experimental valuations are adequate and insightful.

**Weaknesses:**

1. The regret analysis seems to follow quite straightforwardly from Wei et al. (2023) with slight modifications in the hyper-parameter values (as indeed noted on page 17 of the supplementary material, in the description trailing Lemma 18). Therefore, it appears that there is not much novelty in the regret analysis, leaving the novelty to the demonstration of truthfulness and near-optimality of social cost.

2. In continuation to the last sentence of the previous point, there appears to be no motivation to study the criterion delineated in (1). The authors simply proceed to analyse the objective function in (1), simply because it appears in Wei et al. (2023). No further explanation of the criterion in (1) or of its relevance to FL is provided in the paper.

3. The authors consider a very specific form for the valuation function $f(\Delta V_{i,t})$ without motivating the same. Has a similar valuation function been studied in the context of FL? More generally, what conditions must $f$ satisfy for the analyses to go through? These are not discussed in the paper.

4. On the algorithmic side, the authors prove in Lemma 6 that if the selection rule (a rule for selecting a set $S_t$ of clients that would participate in the FL task at time $t$) can be computed in polynomial time, then so can the critical value associated with the selection rule. However, no explicit result stating how much time is taken by Algorithm 1/2 to compute the selection rule is provided in the paper. While the authors provide an explicit scheme to compute the critical value, the authors do not explicitly prove that their Algorithm 1/2 computes the selection rule in polynomial time.

5. The statement of Lemma 7 appears to be contradictory to one of the statements made in its proof (presented in Appendix D). In the proof, the authors identify that $\left(1+\frac{t L^2}{\lambda d} \right)^{-d}$ is a lower bound on a certain ratio of determinants, and further note that "if the hyper-parameter $\beta$ is **greater** than the (preceding) lower bound, it is guaranteed that no client can be essential". However, Lemma 7 states the contrary. Furthermore, the constants $L$ and $\lambda$ appearing in the statement of Lemma 7 are not defined in the main text.

6. There are no comments on the tightness of the bounds on the communication cost and pseudo-regret in relation to the bounds appearing in the work of Wei et al. (2023) (the key piece of work the current paper seems to be based upon). In the process of achieving near-optimal social cost, what is the impact on the communication cost / number of rounds of communication and regret? A formal comparison along these lines with Wei et al. (2023) is missing.

7. Immediately following the statement of Lemma 18 in Appendix E, the authors state that "In each communication round, **all** the clients upload $O(d^2)$ scalars to the server and then download $O(d^2)$ scalars." Why do **all** the clients participate in the communication and not just the ones from the selection set $S_t$ at the communication time instant $t$? This is a little confusing, and not discussed elsewhere in the paper.

The writing of the paper can be significantly improved.

1. The *incentive* $\mathcal{I}_{i,t}$ introduced in (2) does not appear to be used elsewhere in the paper.
2. In Definition 4: $c\_{i}(\mathcal{M}, \widehat{D}\_{-i, t}) \to c\_{i, t}(\mathcal{M}, \widehat{D}\_{-i, t})$.
3. In the paragraph before Lemma 6: "Lemma 5" should be replaced with "Proposition 5".
4. In the proof of Theorem 8: $i \notin S\_t \to i \notin S\_t^\prime$.
5. The notion of "communication threshold" in Theorem 10 is not explained in the paper.

The paper is generally missing the overall feel of an FL paper (no details about the communication protocol, which set of clients participate in communication, what information is communicated from the clients to the server), and seems to only build upon the setting and results of Wei et al. (2023) on the social cost and truthfulness aspects.

**Questions:**

I have discussed the questions under "Weaknesses".

---

> ### Author Response · Authors · 2023-11-17
> **Response to Reviewer 1gFC [Part 1/3]**
>
> We thank the reviewer for the careful review and valuable suggestions to clarify our contribution in comparison with previous works. But we have to first clarify that our focused problem is **federated bandit learning**, not the general federated learning (FL) as the reviewer kept referring to in the review. **We are worried that this might cause most of the reviewer’s misunderstandings about our work (e.g., regarding our model behavior [Q2, Q7, Q9] and novelty [Q1, Q10])**. We ask the reviewer to kindly refer to our general response [CA1] for a detailed discussion on the uniqueness of federated bandit problem, which directly determines how data should be valued in this particular context. More detailed discussions on this aspect are provided in our following responses [A2] and [A3].
>
> Please let us know if you have any further questions, and we are more than happy to incorporate any additional suggestions that could further enhance our work and lead to a more favorable evaluation of our work.
>
> **[Q1]**: It appears that there is not much novelty in the regret analysis, leaving the novelty to the demonstration of truthfulness and near-optimality of social cost.
>
> **[A1]**: As we discussed in the general response [CA4], our method achieves the **same order of regret** as previous works that count on strong assumptions (e.g., the willingness assumption in DisLinUCB [R2], the truthfulness assumption in Inc-FedUCB [R4] as illustrated in the table presented in [CA4]), while we drop all those assumptions. We truly believe our finding that one can achieve the same level of regret while being free from these strong assumptions is already a non-trivial contribution to the community.
>
> We should emphasize that our focus is on devising the incentive mechanism that ensures truthfulness and near-optimal social cost, without compromising the guarantees in regret and communication cost. Moreover, as the previous work has shown the obtained regret and communication upper bounds are already tight [R5], there is little room to develop new regret analysis techniques to improve the results.
>
>
> **[Q2]**:  There appears to be no motivation to study the criterion delineated in (1). No further explanation of the criterion in (1) or of its relevance to FL is provided in the paper.
>
>
> **[A2]**: In our concerned federated bandit problem, optimizing the determinant ratio directly corresponds to the reduction of regret [R4], which is a well-established standard metric in the literature [R1, R2, R3]. Therefore, any feasible solution to Eq. (1) that satisfies the determinant ratio constraint immediately guarantees the near-optimal regret. A more detailed explanation of this criterion is provided in Sections 4.1 and 4.2 in Wei et al. (2023) [R4].
>
> We want to clarify that in our work, using Eq. (1) as our optimization problem is not an assumption but a design choice to achieve our goal. We acknowledge the possibility of other forms of optimization problems that, when solved, can yield similar results with proper proofs.
>
> **[Q3]**: The authors consider a very specific form for the valuation function $f(\Delta V_{i,t})$ without motivating the same. Has a similar valuation function been studied in the context of FL? More generally, what conditions must $f$ satisfy for the analyses to go through? These are not discussed in the paper.
>
> **[A3]**: By the notation $D_{i,t} = f(\Delta V_{i,t})$, we meant to express that our work generalizes the constant setting of data sharing cost, i.e., $D_{i,t} = C_i \cdot \mathbb{I}(\Delta V_{i,t} \neq \mathbf{0})$ as adopted in Wei et al. (2023) [R4].
>
> In fact, none of our analysis across the paper depends on the specific form of $D_{i,t}$ - all that matters is that client $i$ has a value $D_{i,t}$ for its data at time step $t$, and **we do not require any conditions for the function $f$ to satisfy**. To eliminate potential confusion, we will remove the $f(\Delta V_{i,t})$ notation in our final version.

---

> ### Author Response · Authors · 2023-11-17
> **Response to Reviewer 1gFC [Part 2/3]**
>
> **[Q4]**: No explicit result stating how much time is taken by Algorithm 1/2 to compute the selection rule is provided in the paper. While the authors provide an explicit scheme to compute the critical value, the authors do not explicitly prove that their Algorithm 1/2 computes the selection rule in polynomial time.
>
> **[A4]**: For Algorithm 2, the worst-case time complexity of the while loop is $O(N)$. The operation inside the while loop involves finding the maximum element in a set, which takes $O(N)$ time. Therefore, the time complexity of Algorithm 2 is $O(N^2)$.
>
> Let $M$ be the number of iterations of the while loop in Algorithm 1. Hence, the time complexity of Algorithm 1 is $O(M \cdot N^2)$. Specifically, the worst case is to consistently increase the budget $b$ until it reaches the sum of $\widehat D_{i,t}, \forall i \in \widetilde{S}$. Therefore, we can upper bound $M$ by considering the loop-breaking case: $b_0 \cdot (1 + \epsilon)^M \geq \sum_{i=1}^{N} \widehat D_{i,t}$, i.e., $M \leq \left\lceil \log_{1+\epsilon}\left(\sum_{i=1}^{N} \frac{\widehat D_{i,t}}{b_0}\right)\right \rceil$, where $b_0 = \min_{i \in  \widetilde S} \widehat{D}_{i,t}$.
>
> As a result, the overall time complexity of Algorithm 1 is $O( \left\lceil \log_{1+\epsilon}\left(\sum_{i=1}^{N} \frac{\widehat D_{i,t}}{b_0}\right)\right \rceil \cdot N^2)$, which is a polynomial time complexity.
>
> **[Q5]**: A typo in the proof of Lemma 7 (presented in Appendix D). Furthermore, the constants $L$ and $\lambda$ appearing in the statement of Lemma 7 are not defined in the main text.
>
>
> **[A5]**: Thanks for pointing out this typo. We have corrected it (and other typos pointed out by the reviewer) in our updated manuscript. To follow the convention in the bandit literature and reduce the barrier of access, we choose the standard notations $L$ and $\lambda$ to represent the regularization parameters, in line with previous works [R1,R2, R3, R4]. Due to the space limit, we left these standard technical details to the appendix, please refer to the definitions in Lemma 15 and Eq. (12).
>
>
> **[Q6]**: There are no comments on the tightness of the bounds on the communication cost and pseudo-regret in relation to the bounds appearing in the work of Wei et al. (2023). In the process of achieving near-optimal social cost, what is the impact on the communication cost / number of rounds of communication and regret? A formal comparison along these lines with Wei et al. (2023) is missing.
>
>
> **[A6]**: As clarified in [A1], our results on the communication costs and regret are essentially in the same order as those in Wei et al. (2023) [R4], which are already tight order wise. Please see a detailed comparison in the general response [CA4].
>
> As analyzed in our Section 4.3, to achieve a near-optimal social cost, our method needs to slightly relax the determinant ratio constraint by a small constant factor of $(1 - e^{-1})$. Technically, our method guarantees the determinant ratio $\frac{\det(V_{g,t}(S_{t}))}{\det(V_{g,t}(\widetilde{S}))} \geq \beta^{1-e^{-1}}$, while their method guarantees $\frac{\det(V_{g,t}(S_{t}))}{\det(V_{g,t}(\widetilde{S}))} \geq \beta$. Correspondingly, we need to set a lower communication threshold $D_c = \frac{T}{N^2d\log T} - (1-e^{-1}) \sqrt{\frac{T^2}{N^2dR\log T} }\log \beta$ compared to $D_c = \frac{T}{N^2d\log T} - \sqrt{\frac{T^2}{N^2dR\log T} }\log \beta$ in Wei et al. (2023).
>
> Despite these trivial constant gaps, **there is no impact on the order of communication cost and regret**, as we presented in the table of [CA4] and proved in Appendix E.

---

> ### Author Response · Authors · 2023-11-17
> **Response to Reviewer 1gFC [Part 3/3]**
>
> **[Q7]**: Why do all the clients participate in the communication and not just the ones from the selection set $S_t$ at the communication time instant $t$? This is a little confusing, and not discussed elsewhere in the paper.
>
> **[A7]**: To facilitate the data valuation of clients' local updates at time step $t$, all clients need to first upload $\Delta V_{i,t}$ to the server, and only those who decide to participate will subsequently upload the corresponding $\Delta b_{i,t}$. Please refer to [CA3] for a detailed clarification on the communication scheme.
>
> **[Q8]**: The incentive $\mathcal{I}_{i,t}$ introduced in (2) does not appear to be used elsewhere in the paper.
>
> **[A8]**: In the incentivized communication setting, any individual rational client only participates in data sharing if the incentive provided by the server is no less than the client's intrinsic data sharing cost, i.e., $\mathcal I_{i,t} \geq D_{i,t}$. This property is formally introduced in Definition 1 of Wei et al. (2023) [R4], and we have made this notation clearer in our revised manuscript (see Section 3).
>
> **[Q9]**: The notion of "communication threshold" in Theorem 10 is not explained in the paper.
>
> **[A9]**: We have made this notation clearer in our revised manuscript (see Theorem 10). The communication threshold $D_c$ is a hyperparameter of the standard communication event trigger, which is used to control the communication frequency, as defined in the communication protocol (Line 9 of Algorithm 6).
>
> **[Q10]**: The paper is generally missing the overall feel of an FL paper (no details about the communication protocol, which set of clients participate in communication, what information is communicated from the clients to the server), and seems to only build upon the setting and results of Wei et al. (2023) [R4] on the social cost and truthfulness aspects.
>
> **[A10]**: We should clarify that our focus is on mechanism design for incentive compatibility in **federated bandit learning**, instead of the general FL. Please find our detailed clarifications in [CA1] and [CA2].
>
> Due to space limit, we leave the details of the standard communication scheme to Appendix F, where the communication protocol is presented in Algorithm 6. Please see also our general response [CA3].
>
> As we clarified in the general response [CA4], our incentive mechanism design (Algorithm 1) is completely different from Wei et al. (2023) [R4] (Algorithm 3 in their paper), as their design builds on the strong truthfulness assumption. **Note that despite achieving the same results on regret and communication cost bounds, our method requires fewer assumptions but enjoys richer guarantees**.
>
> **References**
>
> [R1]. Yasin Abbasi-Yadkori, Dávid Pál, and Csaba Szepesvári. Improved algorithms for linear stochastic bandits. NeurIPS 2011.
>
> [R2]. Yuanhao Wang, Jiachen Hu, Xiaoyu Chen, and Liwei Wang. Distributed bandit learning: Near-optimal regret with efficient communication. ICLR 2020.
>
> [R3]. Chuanhao Li, and Hongning Wang. Asynchronous Upper Confidence Bound Algorithms for Federated Linear Bandits. AISTATS 2022.
>
> [R4]. Zhepei Wei, Chuanhao Li, Haifeng Xu, and Hongning Wang. Incentivized Communication for Federated Bandits. NeurIPS 2023.
>
> [R5]. Jiafan He, Tianhao Wang, Yifei Min, Quanquan Gu. A Simple and Provably Efficient Algorithm for Asynchronous Federated Contextual Linear Bandits. NeurIPS 2022.

---

> ### Comment · Reviewer_1gFC · 2023-11-20
> **Response to authors' rebuttal (Round 1)**
>
> I thank the authors for their meticulous and comprehensive response. I acknowledge that I erroneously employed the term "federated learning" in my review comments, leading to potential confusion, and I extend my apologies for any misunderstanding. I also wish to acknowledge that I am fully aware of the difference between federated bandit learning and federated learning.
>
> Upon a thorough examination of the authors' detailed rebuttal, I feel that the substantial merit inherent in the paper may not be readily apparent upon initial reading. The key concepts within the paper appear either dispersed or, alternatively, necessitate readers to delve into the specifics of Wei et al.'s (2023) work to fully comprehend the results. I am afraid the reader might miss seeing the paper's significant contributions, primarily due to the writing style.
>
> 1. Given that this is a paper on federated bandit leaning, it is certainly important to mention what rule is followed for communication between the server and the clients, even if this means reproducing a portion of the content from prior works; this is in the interest of making the problem description clear. In their common response [CA3], the authors state that whenever a communication event is triggered, each client uploads $\Delta V_{i,t}$ and, only when incentivised, participates in uploading $\Delta b_{i,t}$. This is an important fact pertaining to the problem setting, and a description of this is missing in Section 3. Even the fact that the communication between the clients and the server takes place on a threshold-based trigger is not made explicit in Section 3. In my opinion, it would be unfair to ask the readers to refer to Appendix F or be aware of the work of Wei et al. (2023) merely to understand this part. It is my humble submission that Section 3 can be rewritten to make the above facts explicit.
>
> 2. Instead of mentioning "Wei et al. (2023) pinpointed the core optimization problem in incentivized communication as follows..." in Section 3 (just before (1)), the authors should perhaps mention that (1) is a "design choice" which they use to model incentivised communication in their work, and that they borrow this from the work of Wei et al. (2023). Again, this is my suggestion to the authors towards improving the writing of the paper to set the objectives clearly in the paper.
>
> 3. I am afraid that the notations $D\_{i,t}$ (true participating cost of client $i$ at time $t$) and $\widehat{D}\_{i,t}$ (the actual reported cost of client $i$ at time $t$) may lead to confusions in understanding the main objective of the paper. The authors state that it is of interest to minimise the social cost which is equal to $\sum_{i \in S_t} D_{i,t}$, whereas the bounds presented in Theorem 9 are for the quantity $\sum\_{i \in S_t} \widehat{D}\_{i,t}$. I think it will be more clearer to the reader if there is an explicit mention that $D\_{i,t}$ is used only in the analysis. Algorithm 6 in Appendix F (the author's algorithm for communication between clients and server), the definition of incentive $\mathcal{I}\_{i,t}$, Eq. (3), and the result in Theorem 9 are all based on the reported cost $\widehat{D}\_{i,t}$ only. It is best to not leave the reader wondering where $D\_{i,t}$ is being used in the paper.
>
> In the interest of saving space, I wish to refrain from pointing out more specifics along the above lines. While I find that the setting and the contributions of the paper are indeed quite interesting (especially regarding the aspects of untruthfulness of clients and social cost, given how important these aspects are from a practical standpoint) and am therefore willing to increase my score to 5 (borderline reject), I find that the writing of the paper can (and perhaps should) be significantly improved. I humbly submit that it would be appropriate for the authors to do a full and thorough rewriting of the paper to delineate the problem description, the important aspects of communication protocol, incentivised communication, and social cost minimisation clearly in their paper, and bring out the contributions more cogently. Also, an inclusion of the table from the common response (comparing the results of the prior works with those of the current paper) in the main text would be apt in my opinion.
>
> **While I am divided between the choice to accept the paper with an essential rewriting for more clarity, and rejecting the paper in the interest of recognising that the authors have very limited time (to do a thorough rewriting) at this stage of the review process, I am inclined to stay with the latter decision**.

---

> ### Author Response · Authors · 2023-11-21
> **Response to Reviewer 1gFC [Round 2, Part 1/3]**
>
> Thank you for the response and acknowledgment of our work's technical contributions. We are particularly glad to find that the reviewer recognizes *“the setting and the contributions of the paper are indeed quite interesting (especially regarding the aspects of untruthfulness of clients and social cost, given how important these aspects are from a practical standpoint)”*.
>
> However, we respectfully disagree with the reviewer’s criticism that “it would be unfair to ask the readers to refer to Appendix F” for a detailed description of the standard communication protocol. Given the strict page limit, we have to make a choice between the details of our technical contributions in this work versus the general background of this line of work. In Section 3 of our original submission, the paragraph below Eq. (1), we explicitly explained that *“A detailed description of this communication protocol is provided in Appendix F”*. And in Appendix F, we carefully described every step in this communication protocol in Algorithm 6 with particular highlights on the uploading and downloading steps (Line 10, 13, 18). Moreover, we have to stress that this communication protocol is quite **standard for federated bandit learning**, almost employed by every work in federated bandits [R4, R5, R6, R7, R8, R9]; and it was actually first introduced in Wang et al. (2020) [R4] paper, not Wei et al. (2023) [R5].
>
> **Given all necessary technical details and background have been presented in our paper and the reviewers found our technical contributions strong, it will be very frustrating to find our paper got rejected, simply because such content was not presented earlier due to space limit**. Moreover, we firmly believe there is no need to rewrite the paper, as all the suggested “confusing notations” and “missing descriptions” have also been explained clearly in the paper. Please refer to the following itemized responses for a detailed justification.
>
> With that being said, from the perspective of disseminating a paper with heavy technical details like ours, we agree it would be more favorable to present the contribution in a way that readers with different backgrounds can grasp it at first glance. To this end, we have included the table in [CA4] in our latest manuscript.
>
> Please let us know if you have any further questions, and we look forward to the possibility of your updated evaluation of our work.
>
> **[Round2-Q1]**: The key concepts within the paper appear either dispersed or, alternatively, necessitate readers to delve into the specifics of Wei et al.'s (2023) work to fully comprehend the results.
>
> **[Round2-A1]**: As we concluded at **the second paragraph in Section 1** of our original submission, our focus is on *“design a truthful incentive mechanism for federated bandits that prevents misreporting while still preserving the nearly optimal regret and communication cost”*, which has clearly stated the expected results and contribution of this work. We have consistently tried to emphasize this in the response to the reviewer’s previous misunderstandings about our contribution in [Q1].
>
> Therefore, we believe we have provided necessary descriptions for the readers to comprehend our results and contributions and there is very little requirement for them to delve into previous works, unless the readers are curious/skeptical about our claim or the general background as in the reviewer’s previous question regarding the optimization problem in [Q2]. What’s more, the discussions regarding those related work have been carefully addressed in our related work section.

---

> > ### Author Response · Authors · 2023-11-21
> > **Response to Reviewer 1gFC [Round 2, Part 2/3]**
> >
> > **[Round2-Q2]**: The communication protocol is an important fact pertaining to the problem setting, and a description of this is missing in Section 3. In my opinion, it would be unfair to ask the readers to refer to Appendix F or be aware of the work of Wei et al. (2023) merely to understand this part. It is my humble submission that Section 3 can be rewritten to make the above facts explicit.
> >
> > **[Round2-A2]**: As stated above, we have explicitly explained this in Section 3 of our original submission, which refers readers to Appendix F where the communication protocol is detailed.
> >
> > Even more noticeably, in the **first sentence of our method section (Section 4.2)**, we reiterated the communication protocol and our distinction to of Wei et al. (2023) again, stating that *“our proposed incentive mechanism Truth-FedBan inherits the incentivized communication protocol by Wei et al. (2023), with the distinction of implementing a truthful incentive search”*. Therefore, we believe the readers should have been well-informed of the communication protocol and can readily find the details in our paper without the need to refer to any external resources.
> >
> > **Disagreements aside, we hope the reviewer would kindly think from the authors’ perspective**: it is crucial to elaborate the innovative contributions in the space-limited main body, which allows us to best share the most exciting findings with the research community. Note that this by no means implies that we overlook the value of discussion about background. In fact, we have already provided very detailed descriptions about the necessary background and reminded the readers in both the preliminary and method sections, albeit we have to leave the detailed content to the appendix due to the space limit.
> >
> > Everyone will be frustrated if we found incremental works are favored over those with significant breakthroughs simply because they have more space in the main body to explain the background, especially in top-tier venues like ICLR.
> >
> > **[Round2-Q3]**: The authors should perhaps mention that (1) is a "design choice" which they use to model incentivised communication in their work, and that they borrow this from the work of Wei et al. (2023). Again, this is my suggestion to the authors towards improving the writing of the paper to set the objectives clearly in the paper.
> >
> > **[Round2-A3]**: First, as presented in the first sentence of the third paragraph of Section 1 of our original submission, our objective is clearly stated: “Following Wei et al. (2023)’s setting for learning contextual linear bandits in a federated environment, **we developed the first incentive-compatible communication protocol Truth-FedBan**”, even before the introduction of the involved optimization problem.
> >
> > Second, regarding the optimization problem in Eq. (1), we acknowledge the possibility of other forms of optimization problems that, when solved, can yield similar results with proper proofs, as explained earlier in the reviewer’s previous question [Q2]. But as we explained before, our formulation has already led to nearly optimal learning performance and desired incentive compatibility. The necessity of studying other optimization problems to realize the same effect seems small. But we are definitely open to other possibilities, which might bring in other aspects of benefit, such as interpretability.
> >
> > More importantly, in the method section (Section 4.2), we have explicitly defined our version of optimization problem Eq. (3), which formally presents our objectives in math, along with additional explanations of the problem design and the relationship between our problem Eq. (3) and Wei et al (2023)’s problem Eq. (1) in the lines right before Eq. (3).
> >
> > Therefore, we believe our current writing of the paper already sets the objectives very clearly for the readers, and there is no need to rewrite the paper as the reviewer suggested.

---

> ### Author Response · Authors · 2023-11-21
> **Response to Reviewer 1gFC [Round 2, Part 3/3]**
>
> **[Round2-Q4]**: I am afraid that the notations $D_{i,t}$ and $\widehat D_{i,t}$ may lead to confusions in understanding the main objective of the paper. The authors state that it is of interest to minimise the social cost which is equal to $\sum_{i \in S_t} D_{i,t}$, whereas the bounds presented in Theorem 9 are for the quantity $\sum_{i \in S_t} \widehat D_{i,t}$. I think it will be more clearer to the reader if there is an explicit mention that $​​D_{i,t}$ is used only in the analysis.
>
> **[Round2-A4]**: We should seriously clarify that the true data sharing cost $D_{i,t}$ of any client $i$ is **NOT** used in any part of our analysis. The reason is simple: the true data sharing costs are private to the clients themselves and are not revealed to the server or any other clients at all, as is standard in truthful mechanism design [R1, R2, R3].
>
> Note that both the notations for social cost (i.e., $\sum_{i \in S_t} D_{i,t}$) and the bound ($\sum_{i \in S_t} \widehat D_{i,t}$) presented in Theorem 9 are **in their intended format**. This is because our Theorem 9 (and other theorems) is narrated from the server’s perspective, and the server only has access to the reported cost $\widehat D_{i,t}$ rather than the true cost $D_{i,t}$. Therefore, it would be incorrect to use the notation $D_{i,t}$ in Theorem 9, as it suggests the server knew the true costs of the clients.
>
> This explains why we need a truthful mechanism to guarantee that clients report $\widehat D_{i,t} = D_{i,t}$; otherwise, the notations would be conflicting with each other. More importantly, with the truthful mechanism design, we can directly minimize the social costs without accessing the private true cost of each client, which is one of the key technical contributions of our work.
>
> **References**
>
> [R1]. Karimireddy, Sai Praneeth, Wenshuo Guo, and Michael I. Jordan. Mechanisms that incentivize data sharing in federated learning. arXiv preprint arXiv:2207.04557 (2022).
>
> [R2]. T. H. Thi Le et al. An Incentive Mechanism for Federated Learning in Wireless Cellular Networks: An Auction Approach. in IEEE Transactions on Wireless Communications, vol. 20, no. 8, pp. 4874-4887, Aug. 2021, doi: 10.1109/TWC.2021.3062708
>
> [R3]. Mu'Alem, Ahuva, and Noam Nisan. Truthful approximation mechanisms for restricted combinatorial auctions. Games and Economic Behavior 64.2 (2008): 612-631.
>
> [R4]. Yuanhao Wang, Jiachen Hu, Xiaoyu Chen, and Liwei Wang. Distributed bandit learning: Near-optimal regret with efficient communication. ICLR 2020.
>
> [R5]. Zhepei Wei, Chuanhao Li, Haifeng Xu, and Hongning Wang. Incentivized Communication for Federated Bandits. NeurIPS 2023.
>
> [R6]. Dubey, Abhimanyu, and AlexSandy Pentland. Differentially-private federated linear bandits. NeurIPS 2020.
>
> [R7]. Chuanhao Li, Huazheng Wang, Mengdi Wang, and Hongning Wang. Communication efficient distributed learning for kernelized contextual bandits. NeurIPS 2022.
>
> [R8]. Chuanhao Li, and Hongning Wang. Asynchronous upper confidence bound algorithms for federated linear bandits. AISTATS 2022.
>
> [R9]. Chuanhao Li, and Hongning Wang. Communication efficient federated learning for generalized linear bandits. NeurIPS 2022.

---

> ### Comment · Reviewer_1gFC · 2023-11-21
> **Response to authors' rebuttal (Round 2)**
>
> I thank the authors for their detailed responses and counter-arguments to my questions. The more I engage with the authors in discussions, the more I am convinced of the soundness of the contributions. That it took so much of explanation from the authors to help me (and perhaps the other reviewers) see the contributions of the paper (in comparison to those of the existing works) more clearly and thereby arrive at a positive evaluation of the paper is of little concern to me. I am only suggesting that the paper be written more carefully so that the key ideas do not remain dispersed all throughout the paper, but are clearly evident in the main text. Somehow, because of the writing style, I feel that the key ideas of the paper and the contributions are dispersed throughout the paper and not explicit in one go, a single reading. Noting that the authors have put in significant efforts to revise their paper heeding to my suggestions and the suggestions of the other reviewers, I am willing to increase my score further. Before I do so, I would like the authors to provide clarifications on the following.
>
> 1. Can the authors clarify if their statement of Lemma 7 is correct? Should $\beta$ be lesser than or greater than the threshold mentioned there for the statement of the Lemma to make sense? The proof of Lemma 7 seems to state the contrary. I believe that $\beta \leq \ldots$ in Lemma 7 should be changed to $\beta \geq \ldots$.
>
> 2. In my understanding, the proposed algorithm guarantees that $\sum_{i \in S_t} \widehat{D}\_{i,t} = \sum_{i \in S_t} D\_{i,t}$ for each time $t$, as the proposed algorithm is shown to satisfy the truthfulness property. Can the authors clarify if my understanding is correct? If so, it may be good to include a statement to this effect after Theorem 9. This would help the reader  correlate the implication of Theorem 9 with the definition of social cost, and thereby reinforce the near-optimality of the social cost achieved by the algorithm. The reason I am nitpicking on this point is because while the algorithm is shown to have a near-optimal value of $\sum_{i \in S_t} \widehat{D}\_{i,t}$ (up to a factor of $1+\epsilon$), it would be incorrect to refer to $\sum_{i \in S_t} \widehat{D}\_{i,t}$ as the social cost, though the Theorem may be stated from the server's perspective.
>
> The authors can perhaps include a sentence immediately after Lemma 6 explicitly stating that their algorithm indeed satisfies the desired polynomial time complexity, and provide a short proof of this in the appendix along the lines of the proof delineated in round 1 of their response.

---

> > ### Author Response · Authors · 2023-11-21
> > **Response to Reviewer 1gFC [Round 3]**
> >
> > We sincerely thank the reviewer again for the timely and constructive suggestions and appreciation of our work! Please let us know if you have any further questions, and we look forward to your updated evaluation of our work.
> >
> > **[Round3-Q1]**: Can the authors clarify if their statement of Lemma 7 is correct? Should $\beta$ be lesser than or greater than the threshold mentioned there for the statement of the Lemma to make sense? The proof of Lemma 7 seems to state the contrary. I believe that $\beta \leq \ldots$ in Lemma 7 should be changed to $\beta \leq \ldots$ .
> >
> > **[Round3-A1]**: We should clarify that both the directions of the inequality ($\beta \leq (1+ tL^2/\lambda d)^{-d}$) in Lemma 7 and that in our proof of Lemma 7 (Appendix D) are **correctly presented as intended**.
> >
> > The reason is that, in our proof we start from the analysis about the root cause of the existence of a monopoly (essential) client. As we explained in Appendix D, a client being essential to the server suggests that even with all the other $N − 1$ clients’ data, the determinant ratio constraint $ \frac{\det(V_{g,t}(S_{t}))}{\det(V_{g,t}(\widetilde{S}))} \geq \beta$ defined in Eq.(3) still cannot be satisfied.
> >
> > Our analysis shows that for any client $i$, the system holds a lower bound (LB) of the determinant ratio when collecting the remaining $N − 1$ clients’ data, i.e., $ \frac{\det(V_{g,t}(S_{t}))}{\det(V_{g,t}(\widetilde{S}))} \geq \text{LB}$.
> >
> > Therefore, by setting the hyper-parameter $\beta$ to be less than the lower bound, we guarantee that no client can be essential, thus eliminating the infinite critical value, as stated in Lemma 7.
> >
> > **[Round3-Q2]**: In my understanding, the proposed algorithm guarantees that $\sum_{i \in S_t} \widehat D_{i,t} = \sum_{i \in S_t} D_{i,t}$ for each time $t$, as the proposed algorithm is shown to satisfy the truthfulness property. Can the authors clarify if my understanding is correct? If so, it may be good to include a statement to this effect after Theorem 9. This would help the reader correlate the implication of Theorem 9 with the definition of social cost, and thereby reinforce the near-optimality of the social cost achieved by the algorithm.
> >
> > **[Round3-A2]**: Yes, the reviewer’s understanding is correct! Thanks for the suggestion, and we have added a discussion after Theorem 9 to help our readers correlate the implication of Theorem 9 with the definition of social cost. Please refer to our latest manuscript for the detailed content.
> >
> > **[Round3-Q3]**: The authors can perhaps include a sentence immediately after Lemma 6 explicitly stating that their algorithm indeed satisfies the desired polynomial time complexity, and provide a short proof of this in the appendix along the lines of the proof delineated in round 1 of their response.
> >
> > **[Round3-A3]**: Thanks for the suggestion, we have included a discussion about the time complexity after Lemma 6, and provided the proof in the appendix. Please refer to our latest manuscript for more details.

---

> > > ### Comment · Reviewer_1gFC · 2023-11-22
> > > **Response to authors' rebuttal (Round 3)**
> > >
> > > I thank the authors for their responses. I think my questions are addressed satisfactorily. I appreciate the authors' effort to revise their manuscript to include the suggestions provided by me and the other authors. I am hereby increasing my score to 8, while also duly recognising that there is room for further improving the writing of the paper to make the contributions stand out better and easily comprehensible in a first reading of the paper.

---

> > > > ### Author Response · Authors · 2023-11-22
> > > >
> > > > We heartily thank the reviewer for his/her dedicated efforts and continuous engagement in the entire review process, which are truly respectful! In the meanwhile, we are also very excited to know that our work is very positively evaluated by the reviewer! This would give us the invaluable opportunity to share our encouraging results and findings with the research community. And we firmly believe that our work will become an important step in the development of federated bandit learning.
> > > >
> > > > As we committed in our earlier responses, we absolutely agree with the reviewer’s suggestions to include more background discussions to the main text that help the readers better comprehend our work and appreciate our unique scientific contributions. We will carefully follow the reviewer's valuable comments and suggestions to best improve the paper’s writing to make our content more easily comprehend for broader audiences.

---

### Author Response · Authors · 2023-11-17
**Common Response to All Reviewers [Part 1/4]**

We sincerely thank all reviewers for their constructive feedback and detailed comments. It is worth noting that there are some shared questions regarding the general research direction of **federated bandit learning**, details about the **communication protocol** of our proposed solution, and confusions regarding **our technical contributions**. Next, we provide our clarifications to these common questions (CQs) in the general response, followed by individual responses to each reviewer's specific questions.

**[CQ1]**: Federated Bandit Learning VS. Federated Learning

**[CA1]**: The bandit problem stands as one of the most important challenges in online machine learning, which has been actively explored and widely applied to sequential decision-making and optimal experiment design applications for decades, owing to its preferable practical performance and sound theoretical guarantees [R1]. For instance, many real-world scenarios involve sequential interactions, such as recommender systems where the system repeatedly interacts with its users to improve its quality of recommendations.

As elaborated in [R2, R3], in federated bandit learning client’s local data is collected on the fly via their interactions with the environment, instead of being static and collected ahead of time in classical federated learning (FL) settings. Specifically,
- Federated Learning: operates on an **offline** setting with fixed datasets, with a sole focus on model estimation.
- Federated Bandit Learning: operates on an **online** setting where data is increasingly collected by the clients’ interactions with the environment, and the distributed clients exchange local data (coordinated by the server) for **improved regret over all clients**.

To summarize, federated bandit learning features two unique challenges compared to classical FL. First, the goal in federated bandit learning is **regret minimization among all clients with low communication cost**, while it is model estimation quality/convergence in FL. Second, information sharing among clients effectively changes clients’ action trajectories and therefore the data to be collected, which causes regret, while in FL the dataset is static over time.

---

> ### Author Response · Authors · 2023-11-17
> **Common Response to All Reviewers [Part 2/4]**
>
> **[CQ2]**: Basic learning objectives and challenges in federated bandit learning.
>
> **[CA2]**: Typically, most existing works on federated bandits before Wei et al. (2023) [R4] assume all clients are altruistic about sharing their information with the server whenever a communication round is triggered. The main interest in this line of research is on designing communication-efficient protocols that balance the following two objectives:
> - **Regret**: the performance of the learning system is measured by the cumulative regret over all $N$ clients in the finite time horizon $T$, i.e., $R_{T}=\sum_{t=1}^{T} r_{t}$, where $r_t = \max_{ \mathbf x \in \mathcal A_t} \mathbf{E}[y|\mathbf x] - \mathbf{E}[y_t|\mathbf x_t]$ is the regret incurred by client $i_{t}$ at time step $t$. Intuitively, regret measures the difference to the cumulative reward that the client should have collected in the hindsight.
> - **Communication cost**: under the federated learning setting, the system also needs to keep the communication cost $C_{T}$ low, which is measured by the total number of scalars being transferred across the system up to time $T$.
>
> With the linear reward assumption, i.e., $f(\mathbf x)=\mathbf x^{\top}\theta_{\star}$, where $\theta_{\star}$ denotes the unknown model parameter. In the centralized setting where the server has access to all client’s local data, a ridge regression estimator $\hat \theta_{t} = V_{t}^{-1} b_{t}$ can be constructed based on sufficient statistics from all $N$ clients at each time step $t$, where $V_{t}=\sum_{s=1}^{t} \mathbf x_{s} \mathbf x_{s}^{\top}$ and $b_{t}=\sum_{s=1}^{t} \mathbf x_{s} y_{s}$. Using $\hat \theta_t$ with the standard upper confidence bound (UCB)-based arm selection strategy [R5], one can obtain the optimal regret $R_{T}=O(d\sqrt{T})$.
>
> To achieve this optimal regret bound in the federated setting, a naive method is to immediately share each client’s local updates to all other clients in the system, which essentially recovers its centralized counterpart. However, this solution incurs a disastrous communication cost $C_{T}=O(d^2NT)$. On the other extreme, if no communication occurs throughout the entire time horizon (i.e., $C_{T}=0$), the regret upper bound can be up to $R_{T}=O(d\sqrt{NT})$. This creates the tension between minimization of regret and communication cost.
>
> To balance this trade-off, recent studies have introduced communication-efficient protocols for federated bandits where clients only occasionally communicate with the server, controlled by the communication event trigger [R2]. Wei et al. (2023) [R4] brought up the need of incentivized communication into federated bandit learning, which suggests a client shares its local update with the server only when the incentive it would receive from the server outweigh its cost of sharing. However, **their proposed incentive mechanism is built on a naive assumption that all clients truthfully report their cost**. This clearly undermines their solution’s practicality and our goal is to design the mechanism that being truthful is the best response of every client, i.e., incentive compatibility. And this is the general background and motivation of our research in this paper.

---

> > ### Author Response · Authors · 2023-11-17
> > **Common Response to All Reviewers [Part 3/4]**
> >
> > **[CQ3]**: Communication protocol in the context of **incentivized** federated bandits learning.
> >
> > **[CA3]**: As detailed in our Appendix F (Algorithm 6), at each time step $t$, an arbitrary client $i_t$ becomes active and interacts with the environment to collect new data $(x_t, y_t)$ based on the standard upper confidence bound (UCB) strategy (Line 6), and then updates its sufficient statistics $V_{i_t,t}$ and $b_{i_t,t}$ and local updates $\Delta V_{i_t,t}$ and $\Delta b_{i_t,t}$.
> >
> > Whenever a communication event (Line 9) is triggered, **all** clients upload their latest sufficient statistics update $\Delta V_{i,t}$  to the server (Line 10) to facilitate data valuation and participant selection (Line 11). Note that this disclosure does not compromise clients’ privacy, as the clients’ secret lies in $ \Delta b_{i,t}$ that is constructed by their collected rewards. Only participating clients will upload their $\Delta b_{i,t}$ to the server (Line 13); and a client decides to upload $\Delta b_{i,t}$ only when the server-provided incentive is no less than their claimed cost (i.e., individual rationality [R4, R6]). After collecting data from all participants, the server downloads the aggregated updates $\Delta V_{-i,t}$ and $ \Delta b_{i,t}$ to every client i (Line 17-20).
> >
> > Note that in the context of incentivized federated bandits, even using the standard UCB-based arm selection strategy [R5] and commutation-efficient event trigger [R2] is not enough to guarantee near-optimal learning performance, as clients only participate in data sharing when properly incentivized. **Therefore, the main interest in this line of research is the design of an incentive mechanism that aims to solve the optimization problem of Eq.(1), as pointed out by [R4]**. Our work goes even further - we not only solve the optimization problem with a better guarantee on the objective, but also ensure the incentive compatibility of the mechanism, i.e., we do **not** need to assume the clients are truthful to begin with.

---

> > > ### Author Response · Authors · 2023-11-17
> > > **Common Response to All Reviewers [Part 4/4]**
> > >
> > > **[CQ4]**: Contribution compared to Wei et al. (2023).
> > >
> > > **[CA4]**: Compared to the previous work by Wei et al. (2023) [R4], we share the same communication framework (Algorithm 6) along with the standard UCB-based arm selection strategy [R5] and communication event trigger [R2]. We acknowledge that Wei et al. (2023) [R4] is the first work to introduce the incentivized communication problem for federated bandits and present a plausible solution to the optimization problem. But as we explained in our submission, they naively assumed all clients report their true data sharing cost, which makes their solution vulnerable to manipulations by malicious clients. For example, clients can arbitrarily raise their claimed data-sharing cost to increase their received incentives from the server. And there is no way in [R4]’s solution to prevent this from happening.
> > >
> > > In contrast, we do not assume truthfulness; instead, we design the mechanism to ensure truthfulness, while maintaining near-optimal bandit learning performance. To demonstrate the contribution of our work, we provide a comprehensive comparison among the most related works as follows.
> > >
> > > | Methods | Regret | Communication Cost | Individual Rationality | Incentive Compatiblity | Social Cost Near-Optimality |
> > > |:------------------:|:------:|:-------------------:|:---------------------:|:-----------------------:|:------------------------:|
> > > | DisLinUCB [R2] | $O(d\sqrt{T} \log T)$ | $O(N^{2} d^3\log T)$ | ❌ | ❌ | ❌ |
> > > | Inc-FedUCB [R4] | $O(d\sqrt{T} \log T)$ | $O(N^{2} d^3\log T)$ | ✅ | ❌ | ❌ |
> > > | Truth-FedBan (Ours)| $O(d\sqrt{T} \log T)$ | $O(N^{2} d^3\log T)$ | ✅ | ✅ | ✅ |
> > >
> > >
> > > As presented in the table, compared to the seminal work [R2] in federated bandits, Wei et al. (2023) [R4] first relaxed the assumption on the willingness of clients in data sharing and devised the first incentivized communication protocol that achieves near-optimal regret and communication cost. Our work goes beyond [R4] by further relaxing the truthfulness assumption, and introducing a novel incentive mechanism that guarantees both incentive compatibility and social cost optimality while maintaining the near-optimal learning performance.
> > >
> > > **To summarize, the core conceptual contribution of this paper is, for the first time, we demonstrate the possibility of simultaneously achieving incentive compatibility and optimal regret in federated bandit learning.** The realization of this possibility is highly non-trivial and results from our novel mechanism design, which involves carefully selecting a subset of participants at every round with guaranteed properties. On the technical side, the main challenge lies in proving the incentive compatibility property, which essentially relies on the monotonicity of the proposed Algorithm 1. Interestingly, despite the widespread use of such greedy algorithms in submodular maximization [R7], this monotonicity result is unknown in previous literature to the best of our knowledge. We believe this additional contribution is of independent interest to broader fields beyond the bandit research community.
> > >
> > >
> > > **References**
> > >
> > > [R1]. Tor Lattimore and Csaba Szepesvari. Bandit algorithms. Cambridge University Press, 2020.
> > >
> > > [R2]. Yuanhao Wang, Jiachen Hu, Xiaoyu Chen, and Liwei Wang. Distributed bandit learning: Near-optimal regret with efficient communication. ICLR 2020.
> > >
> > > [R3]. Chuanhao Li and Hongning Wang. Asynchronous upper confidence bound algorithms for federated linear bandits. International Conference on Artificial Intelligence and Statistics. PMLR, 2022.
> > >
> > > [R4]. Zhepei Wei, Chuanhao Li, Haifeng Xu, and Hongning Wang. Incentivized Communication for Federated Bandits. NeurIPS 2023.
> > >
> > > [R5]. Yasin Abbasi-Yadkori, Dávid Pál, and Csaba Szepesvári. Improved algorithms for linear stochastic bandits. NeurIPS 2011.
> > >
> > > [R6]. Roger B Myerson and Mark A Satterthwaite. Efficient mechanisms for bilateral trading. Journal of economic theory, 29(2):265–281, 1983.
> > >
> > > [R7]. Rishabh K. Iyer and Jeff A. Bilmes. Submodular optimization with submodular cover and submodular knapsack constraints. NeurIPS 2013.

---

### Author Response · Authors · 2023-11-19
**A Gentle Reminder to the Reviewers**

Dear reviewers,

We hope this message finds you well. We understand that the author reviewer discussion is a critical component of the ICLR review process, and we would like to remind you that the rebuttal period is scheduled to **conclude on November 22nd**. Given there are only 4 days left before the deadline, we would like to call for our reviewers' attention to our provided responses.

We are fully committed to engaging with the discussions, if any further information or clarification is needed regarding our response. Thank you for your time and attention to this matter! Your efforts in reviewing submissions are deeply appreciated.

Best,

Authors

---

> ### Comment · Area_Chair_7AtW · 2023-11-19
> **Request to reviewers to respond to authors' comments**
>
> Dear reviewers,
>
> The authors have provided detailed replies to your comments. Please go through them and at least acknowledge that you've read through them regardless of whether or not you change your score.
>
> Regards,
>
> Your AC

---

### Meta-Review · Area_Chair_7AtW · 2023-12-05

**Metareview:**

The setting studied in this paper is rather interesting and highly unique, incorporating federated bandits with incentivized truthful communications. There were many spirited discussions between the authors and reviewers, which is what the system intended. After the discussions between authors and reviewers, the reviewers came to a consensus that there is certainly a lot of value in the paper, though the exposition can be improved. I trust that the authors will take the feedback into account in the preparation of the final version of this paper.

**Justification For Why Not Higher Score:**

The exposition can be improved to properly situate the paper in the vast literature on bandits and federated bandits. Some of the comments by the reviewers should be incorporated into the final version of the paper.

**Justification For Why Not Lower Score:**

There is clearly a lot of merit in the paper. Hence the score is not low.

---

### Decision · Program_Chairs · 2024-01-16

Accept (poster)